# MANIBOX: ENHANCING SPATIAL GRASPING GENERALIZATION VIA SCALABLE SIMULATION DATA GENERATION

## ABSTRACT

Learning a precise robotic grasping policy is crucial for embodied agents operating in complex real-world manipulation tasks. Despite significant advancements, most models still struggle with accurate spatial positioning of objects to be grasped. We first show that this spatial generalization challenge stems primarily from the extensive data requirements for adequate spatial understanding. However, collecting such data with real robots is prohibitively expensive, and relying on simulation data often leads to visual generalization gaps upon deployment. To overcome these challenges, we then focus on state-based policy generalization and present **ManiBox**, a novel bounding-box-guided manipulation method built on a simulation-based teacher-student framework. The teacher policy efficiently generates scalable simulation data using bounding boxes, which are proven to uniquely determine the objects' spatial positions. The student policy then utilizes these low-dimensional spatial states to enable zero-shot transfer to real robots. Through comprehensive evaluations in simulated and real-world environments, ManiBox demonstrates a marked improvement in spatial grasping generalization and adaptability to diverse objects and backgrounds. Further, our empirical study into scaling laws for policy performance indicates that spatial volume generalization scales positively with data volume. For a certain level of spatial volume, the success rate of grasping empirically follows Michaelis-Menten kinetics relative to data volume, showing a saturation effect as data increases.[1]

## 1 INTRODUCTION

Robotic manipulation in dynamic environments is pivotal for advancing modern robotics (Fu et al., 2024). Equipping robots with precise grasping capabilities in unstructured settings not only extends their utility beyond traditional static scenarios but also enhances their real-world effectiveness, especially in industrial and domestic settings. A robot policy that achieves **spatial generalization**—*defined as the ability of a manipulation model to complete tasks regardless of the target object's position within a defined spatial volume*—would be particularly practical. Such generalization is vital when deploying robots in diverse environments, such as different households, where they must handle various objects on different platforms. The capability to perform successfully across these varied spatial conditions is essential, as objects in real-world settings are rarely fixed in place, and effective manipulation demands adaptability to a broad range of spatial challenges.

Recent works have made notable progress in robotic manipulation in multi-task and multi-environment scenarios, which is largely driven by end-to-end training methodologies (Ahn et al., 2022; Shridhar et al., 2022; 2023; Zitkovich et al., 2023). Notably, some large embodied foundation models for manipulation, such as those fine-tuned from Vision-Language Models (VLMs) for multi-task robot manipulation tasks (Brohan et al., 2022; Kim et al., 2024), have made significant strides. However, these large manipulation models derived from VLMs often lack sufficient spatial awareness and reasoning capabilities, posing substantial challenges for large-scale spatial generalization in real-world applications (Cheng et al., 2024a). Our analysis in Lemma 1 suggests that achieving spatial generalization over larger volumes requires substantially more data, a relationship we further empirically verify to follow a positive scaling law (see Figure 4). For example, training a model to

---

[1]Our data and code are available in the supplementary material.

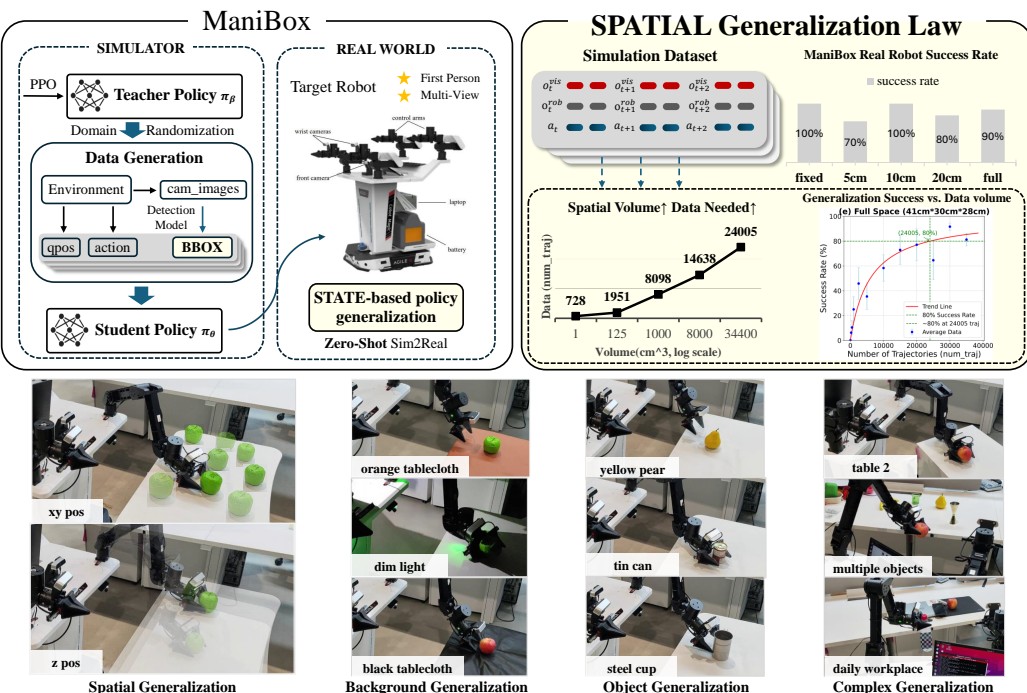

Figure 1: **Overview.** We introduce ManiBox, a bbox-guided manipulation method using a teacher-student framework to enhance spatial generalization in grasping policies. We reveal that spatial volume generalization scales positively with data volume, with grasping success following Michaelis-Menten kinetics relative to data volume for specified spatial volumes. Extensive real-world tests show ManiBox's robust adaptability to varied spatial positions, objects, and backgrounds.

grasp at a fixed point may require only 50 to 600 trajectory datasets. However, when generalizing the model to operate within a spatial volume of $34,400cm^3$ ($41cm \times 30cm \times 28cm$), which approximates the maximum reach of our robotic arm, the data requirement increases by roughly 40-fold, demanding tens of thousands of trajectories

However, collecting such a large volume of data on real robots is prohibitively expensive. For instance, RT-1 required 17 months to accumulate 130K episodes (Brohan et al., 2022), which is a level of resource commitment unfeasible for most research labs. Although simulation data can help bridge this gap, significant high-dimensional visual discrepancies between simulated and real environments, known as the Sim2Real gap, pose additional challenges. While prior efforts have aimed to enhance performance using various visual generalization techniques, each method comes with its drawbacks. Some demand unobstructed third-person perspectives (Yuan et al., 2024), some consume substantial computational resources (Shridhar et al., 2023; Ahn et al., 2022). These requirements can impede real-time processing and practical deployment on resource-limited robotic platforms.

While perception models are essential for object and environmental recognition, the true effectiveness of a robot in dynamic settings hinges on the robustness of its policy execution. A well-generalized policy empowers the robot to autonomously adjust its actions in real-time, effectively compensating for visual inaccuracies and adapting to changes in object positions or environmental dynamics. Such adaptability is crucial for achieving task success and maintaining system stability, especially in the face of imperfect or noisy perception data. To further enhance the robot's ability to manage unpredictable scenarios (e.g. Sec. B.2) and ensure reliable real-world performance, we shift our focus to **state-based policy generalization** instead of relying solely on vision-based methods. Our Lemma 2 demonstrates that bounding boxes, when captured from multiple cameras, effectively encapsulate the 3D structure of convex objects, providing an optimal low-dimensional state for policy generalization. Building on this foundation, we introduce a novel bounding-box-guided methodology **ManiBox** to enhance spatial generalization and adaptability.

In the simulator, ManiBox utilizes reinforcement learning (RL) with privileged information to train a sophisticated teacher policy. This policy generates scalable robot trajectory data, replacing tra-

ditional visual inputs with bounding box coordinates to facilitate spatial generalization of grasping policies. Subsequently, the student policy, trained on this simulation data, achieves robust zero-shot transfer to real-world tasks, leveraging bounding boxes identified by advanced open-vocabulary detection models, such as YOLO-World (Cheng et al., 2024b). Extensive experiments with both simulators and real robots demonstrate a direct correlation between data volume and spatial generalization performance: *more data consistently leads to higher grasping success rates, and generalizing to larger spatial volumes requires even more data.* For instance, to reliably grasp objects within a $1000cm^3$ workspace, tens of thousands of trajectories from various positions are required. ManiBox achieves nearly perfect success rates in real-world grasping tasks by leveraging large-scale simulation data generated across diverse spatial volumes. Furthermore, ManiBox exhibits robust generalization, successfully grasping objects in new scenarios, including unfamiliar object types, complex surfaces, changing lighting conditions, and environments with distractions.

Our contributions include: (1) We introduce **ManiBox**, an innovative bounding-box-guided manipulation method that addresses the limitations of visual generalization, offering a robust solution for spatial generalization in grasping tasks; (2) ManiBox significantly enhances adaptability to diverse spatial positions, object types, and backgrounds, as demonstrated by extensive experiments in both simulated and real-world environments; (3) Through scalable simulation data generation, we provide novel insights into the relationship between data volume and spatial generalization, establishing a framework for improving generalization across a variety of embodied systems.

## 2   RELATED WORK

**Learning from Simulation.**    Simulators provide an efficient and scalable way to expand embodiment datasets, offering a cost-effective means for parallel data collection and faster iteration cycles. There are three main approaches for leveraging simulation in robotic control: 1) training reinforcement learning (RL) agents through interaction within the simulator (Huang et al., 2023; Radosavovic et al., 2024; Yuan et al., 2024), 2) using imitation learning based on expert demonstrations generated in simulation (Xie et al., 2020; Tan et al., 2024), and 3) employing a teacher-student framework, where privileged information from simulation trains a teacher policy to guide a student policy (Lee et al., 2020a; Geng et al., 2023; Zhuang et al., 2023), as demonstrated in the ManiBox framework. These methods reduce the reliance on expensive real-world data collection.

Various techniques have been developed to bridge the Sim2Real gap, including realistic simulators (Todorov et al., 2012; James et al., 2020; Makoviychuk et al., 2021), domain randomization (Tobin et al., 2017; Peng et al., 2018; Zhuang et al., 2023), and vision-based policy generalization (Geng et al., 2023; Tan et al., 2024; Ying et al., 2024; Yuan et al., 2024). These approaches enable robots to generalize across diverse visual and physical conditions, which are critical in real-world environments. For example, domain randomization allows policies trained in varied simulated conditions to transfer effectively to real-world scenarios (Tobin et al., 2017; Peng et al., 2018). While real-world datasets such as Open X-Embodiment (Padalkar et al., 2023) provide 1M demonstrations, they are much smaller than datasets in other domains like language and images (Lehmann et al., 2015; Mühleisen & Bizer, 2012; Weyand et al., 2020; Wu et al., 2019). Thus, simulation-generated datasets play a vital role in compensating for the limitations of real-world data collection.

**Visual Generalization for Robotic Manipulation.**    Visual generalization is fundamental for robotic manipulation in dynamic environments, where robots must handle varying visual inputs and changing conditions, such as object positions and lighting variations. Vision-based policy generalization, which leverages both simulated and real-world data, addresses these challenges. By simulating diverse visual scenarios, robots can better generalize to new environments and tasks. For instance, multi-task and multi-object manipulation studies have extended behavior cloning to handle a wide range of visual inputs, improving generalization to unseen objects and scenes (Vuong et al., 2023; Fang et al., 2023). Additionally, language-conditioned policies (Brohan et al., 2022; Zitkovich et al., 2023; Shridhar et al., 2023) and imitation learning from human videos (Chen et al., 2021; Nair et al., 2023) enable robots to interpret and manipulate objects using both visual and language cues.

However, scaling real-world demonstrations to improve generalization remains challenging due to resource constraints. Simulation-based learning mitigates this by offering large, diverse visual data. Domain randomization techniques (Tobin et al., 2017; Peng et al., 2018) have proven effective in generating varied visual conditions, enabling models to generalize more robustly across real-world

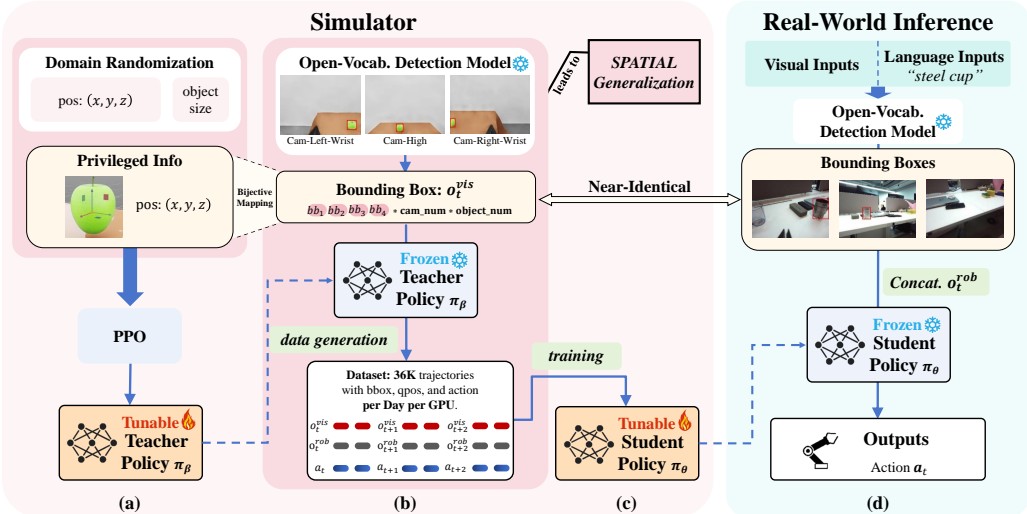

Figure 2: **ManiBox illustration.** (a) Utilizing PPO and domain randomization, we train a sophisticated teacher policy in the simulator that utilizes privileged object information to determine actions. (b) The teacher policy generates scalable trajectory data, utilizing bounding box coordinates instead of traditional high-dimensional visual inputs or privileged information. (c) A state-based student policy, generalizable and capable of zero-shot transfer, is trained on this extensive simulation dataset, greatly improving spatial generalization. (d) Guided by bounding boxes, the student policy precisely executes actions for real robots, achieving improved generalization capabilities.

tasks. While methods like bounding box annotations assist in specific tasks such as pose estimation (Huang et al., 2023), understanding the relationship between dataset scale and generalization remains an area that requires further exploration. As simulation data grows, more research is needed to evaluate how data volume impacts policy generalization, particularly in dynamic environments.

## 3 METHODOLOGY

In this section, we first formulate the problem and then introduce our method ManiBox, including the teacher policy training, simulation data generalization, and student policy distillation.

### 3.1 PROBLEM FORMULATION

For robotic manipulation, especially within dynamic environments, achieving spatial generalization (i.e., the ability to effectively perform tasks across varied spatial configurations) is essential for usability and overall efficacy. We formulate robotic manipulation as a partially observable Markov decision process (POMDP) (Cassandra et al., 1994), by considering the robot's limited access to complete state information, the necessity to make decisions under uncertainty, as well as the inherently stochastic nature of its interactions with the environment. Formally, the decision-making process is characterized by the tuple $\mathcal{M} = (\mathcal{S}, \mathcal{A}, \mathcal{O}, \mathcal{T}, \mathcal{R}, \gamma)$, where $\mathcal{S}$ represents the state space, including all configurations of the robot and its environment; $\mathcal{A}$ denotes the action space, containing all possible robot actions; $\mathcal{O}$ is the observation space, which reflects the limited state information accessible to the robot; $\mathcal{T}$ defines the state transition dynamics, $\mathcal{T}(s_{t+1}|s_t, a_t)$; $\mathcal{R}$ is the reward function, providing feedback based on the state-action pair $\mathcal{R}(s_t, a_t)$; and $\gamma$ is the discount factor.

In real-world scenarios, the robot receives partial observations $o_t \in \mathcal{O}$ through an observation function, which provides incomplete and potentially noisy information about the true state $s_t$. The observation can be represented as $o_t \sim \mathcal{O}(s_t)$. The robot's objective is to find a policy $\pi$ that maps observations (or histories of observations) to actions $a_t \sim \pi(a_t|\tau_t)$, where $\tau_t = (o_0, a_0, o_1, a_1, \ldots, o_t)$ is the history up to time $t$. The policy is expected to maximize the expected cumulative reward:

$$J(\pi) = \mathbb{E}_\pi \left[ \sum_{t=0}^{\infty} \gamma^t \mathcal{R}(s_t, a_t) \right], \tag{1}$$

where the expectation is over the trajectories induced by the policy $\pi$. However, solving POMDPs for manipulation tasks is computationally expensive, as it requires maintaining a belief distribution

over all possible states, including the precise position, orientation, and object dynamics. This complexity is compounded by the need to handle occlusions, sensor noise, and dynamic changes in the environment, making it challenging for robots to operate efficiently in real-world settings where the state is constantly evolving and uncertain (Kaelbling et al., 1998; Kochenderfer, 2015).

To reduce this complexity and ensure Sim2Real transfer, we adopt a widely used teacher-student framework (Lee et al., 2020a; Geng et al., 2023), building upon which, we develop **ManiBox**, a novel bounding-box-guided manipulation method that offers significant advantages in spatial generalization over vision-based RL methods. First, ManiBox enables efficient, scalable data generation, as the teacher policy in simulation can produce a large and diverse set of trajectories. Additionally, by using low-dimensional bounding box states, the state-based student policy can accelerate learning and effectively transfer from simulation to the real world.

However, this framework also faces notable challenges. The teacher policy must efficiently explore a high-dimensional state space in simulation, which can be computationally intensive. The student policy must derive optimal actions from incomplete observations and adjust to real-world variations such as sensor noise and dynamic environmental conditions not present in simulations. Detailed strategies for overcoming these challenges are discussed in the following sections.

## 3.2 TEACHER POLICY TRAINING

To automatically and efficiently generate a large amounts of diverse data, we first train an expert teacher policy $\pi_\beta : \mathcal{S} \to \mathcal{A}$ using RL, as the expert data generator. The **teacher policy** $\pi_\beta(\boldsymbol{a}_t|\boldsymbol{s}_t)$ is trained in a simulated environment where it has access to full state information $\boldsymbol{s}_t$, including privileged information such as precise object positions. The teacher policy is optimized using reinforcement learning (RL) to maximize the expected reward:

$$J_\beta = \mathbb{E}_{\pi_\beta} \left[ \sum_{t=0}^\infty \gamma^t \mathcal{R}(\boldsymbol{s}_t, \boldsymbol{a}_t) \right], \tag{2}$$

where the reward function is defined to guide the policy in grasping the object and moving it to a specified location. We select the grasping task to validate our approach's core concept. This allows the teacher to generate a diverse set of optimal trajectories across a wide range of spatial configurations. These trajectories serve as training data for the student policy, guiding it to learn effective manipulation policies.

The advantages of state-based teacher policies lie in their lower-dimensional states, enabling more efficient learning, reduced exploration space through privileged information, and support for larger parallel environments. To leverage these benefits, in addition to optimizing the policy using the Proximal Policy Optimization (PPO) (Schulman et al., 2017) algorithm, we make adjustments to the RL environment to reduce the exploration space and make it easier for the policy $\pi_\beta$ to find the point with the highest return. Moreover, to support follow-up Sim2Real and spatial generalization, we carefully design the environment below (More details are in Appendix E.1):

**Environment Setup.** To ensure efficient data generation and policy training, we utilize Isaac Lab (Mittal et al., 2023), which supports large-scale parallelism environments. The scene includes basic elements like a platform, objects, and a robot, arranged simply to facilitate future domain randomization. The robot's goal is to grasp the object on the table and move it to a designated location. We integrate the dual-arm Mobile ALOHA (Fu et al., 2024) robot, equipped with three first-person view RGB cameras, into the simulator and ensure its dynamics are consistent with the real world.

**State.** For training efficiency, our teacher policy is state-based and can access full state information $\boldsymbol{s}_t$. As the robot lacks visual input, we include privileged information like the object's position in the state information. The privileged information accelerates RL training and enables the agent to learn more efficient and generalized policies.

**Domain Randomization.** To improve the generalization of the teacher policy and enhance the spatial diversity of the generated data, we adopt several domain randomization techniques:
- *objective position* $(x, y, z)$: To generalize the grasping policy to the whole space, we apply the domain randomization for all three dimensions of $x$, $y$, and $z$, ensuring that the $x$-coordinate and $y$-coordinate of the object, which is chosen randomly, can cover a large range. Moreover, the height of the platform is randomized to ensure that the $z$-coordinate of the object in the data can cover the feasible working space of the robotic arm.

- *object size*: To ensure that the grasping policy covers objects of *various sizes*, the size of the object is also randomized.

These two domain randomization techniques allow teacher policy can grasp different objects of different positions and sizes, which is significant for real-world generalization. By combining privileged information and domain randomization, the teacher policy can efficiently generate a large amount of diverse data to achieve spatial generalization.

## 3.3 SIMULATION DATA GENERATION

We utilize the teacher policy to automatically generate data for training a student policy that can be deployed in the real world across different objects, grasping spatial positions, and backgrounds.

**Data Volume Estimation.** To guide the data volume needed for policy generalization to a specific spatial volume, we now analyze the sample efficiency for generalizing in a $b \times b \times b$ cube. Based on the classical learning theory, the sample complexity of uniform convergence property is in proportion to the VC dimension of the model hypothesis set (Shalev-Shwartz & Ben-David, 2014). Extending previous results about generalization bounds for multi-task learning (Crammer & Mansour, 2012), we can have the following results:

**Lemma 1** (Details are in Appendix A.1). *For training a grasp agent that can generalize spatially within a cube whose size is $b \times b \times b$ via imitation learning, under some mild assumptions in Crammer & Mansour (2012), the VC dimension is at most proportional to $d \log \left(C b^3 d\right)$, where $d$ is the VC dimension of the hypothesis set for grasping the fixed point and $C$ is a constant independent of $d, b$.*

This lemma demonstrates that the data required for policy generalization to a certain volume grows, thus it is necessary to collect data from the simulator. This result gives us a rough estimation of how much data we need to collect with the increasing of the spatial cube needs to generalize.

In light of this Lemma 1, we need a large scale of trajectories to train a student policy feasible under full spatial range. This calls for conciseness in data generation and collection, otherwise it will place a heavy burden on student policy training. On the other hand, since the teacher policy trained with privileged information (i.e. object positions) cannot directly transfer from simulation to reality (Sim2Real), the student policy needs visual observations that implicitly encode object position information. However, there remains a significant Sim2Real gap between visual images in the simulator and those in the real world because the vast complexity of real-world environments cannot be fully simulated by the simulator (e.g. various backgrounds and objects). Additionally, vision-based policy is difficult to train on a large scale and requires data augmentation.

To address this Sim2Real gap and enable robust spatial generalization, we propose using an individual vision module to extract the bounding box from camera views as a **consistent, low-dimensional, and complete** representation of the object's spatial properties, such as position and size.

**Using Bounding Boxes to Provide Complete 3D Information about the Object.** To efficiently generate a large amount of simulator data that aligns with real-world visual images, the vision module must be robust, mature, and efficient. Considering open-vocabulary detection, we selected YOLO-World (Cheng et al., 2024b) as the frozen 2D visual feature extraction module. This provides us with the bounding box of the objects as the 2D low-dimensional visual feature, which is consistent and low-dimensional. Notably, the inference time of YOLO-World for a single image is around 30ms , ensuring the data generalization efficiency and real-time inference in the real world.

In addition, to ensure that these multi-view 2D low-dimensional visual features can offer complete information about object position and size, we present a lemma: Suppose the object is a sphere, the 3D information of the object can be uniquely determined by the bounding boxes from two cameras:

**Lemma 2** (Details and proofs are in Appendix A.2). *Given: (1) Two pinhole cameras with known intrinsic parameters $K_i$ and extrinsic parameters $(R_i, t_i)$; (2) A sphere of unknown radius $r$ is fully visible in both camera images without occlusion; (3) Normalized image coordinates of the bounding boxes of the sphere in both images: Camera $i$: $(u_{1min}, v_{imin}, u_{imax}, v_{imax})$, the image dimensions are known (width $W_i$ and height $H_i$ for each camera).*

*Then the **3D coordinates of the sphere's center** C **in the world coordinate system and the radius** $r$ **of the sphere can be uniquely determined** using the known camera parameters, and the normalized bounding box coordinates, provided that the cameras are not in degenerate configurations.*

This lemma illustrates that 3D position information of a sphere can be implied by utilizing multi-view bounding boxes (Hartley & Zisserman, 2003), indicating our ManiBox can capture complete spatial position information of target objects for real-world control policies.

In short, our approach of using an individual vision module simplifies the transition from simulation to the real world by providing the student policy with consistent inputs across both domains. Specifically, the student policy receives both proprioceptive data $o_t^{\text{rob}}$ and bounding box data $o_t^{\text{vis}}$ as inputs:

$$\pi_\theta : \mathcal{O} = \mathcal{O}_{\text{rob}} \times \mathcal{O}_{\text{vis}} \to \mathcal{A}. \tag{3}$$

### 3.4 STUDENT POLICY DISTILLATION

While the teacher policy and data generation are completed in the simulator, the student policy is deployed on a real robot and relies only on partial observations in the real world. The student policy needs to ensure high generalizability in real-world environments. Consequently, real-world variations, such as sensor noise and dynamic environments, pose challenges not encountered in the simulation environment to our student policy. For instance, an open-vocabulary detection model(namely YOLO-World (Cheng et al., 2024b) in our practice) can fail to recognize real-world target objects at times because of background distraction or robot gripper occlusion, while the recognition of the simulated target object rarely fails.

In this context, we introduce the random mask to bridge this sim-to-real gap and further improve the robustness of our policy. In detail, when distilling our student policy from the simulation data $\mathcal{D}$, for each trajectory, we randomly mask some bboxes(i.e., set them to 0) with a probability predefined as `random_mask_ratio`, indicating that this camera does not recognize the target object. This approach replicates the real-world scenarios where the open-vocabulary detection model fails to generate the bounding box.

On top of this expected unpredictable failure and partial observability, our policy requires encoding sufficient history information (Hafner et al., 2019; Lee et al., 2020b; Ying et al., 2023). For this reason, we use RNN for our student policy:

$$\begin{aligned}
\boldsymbol{x}_t, \boldsymbol{h}_t &= \text{LSTM}(\boldsymbol{s}_t', \boldsymbol{h}_{t-1}), \\
\boldsymbol{a}_t &= \text{FC}_2(\text{drop}(\text{gelu}(\text{FC}_1(\boldsymbol{x}_t)))),
\end{aligned} \tag{4}$$

where $\boldsymbol{s}_t'$ and $\boldsymbol{h}_t$ denotes the input state and the hidden state of the LSTM at time $t$, $\boldsymbol{x}_t$ is the intermediate output of the LSTM, and $\boldsymbol{a}_t$ is the action vector output by the policy. Furthermore, $\text{LSTM}(\cdot), \text{gelu}(\cdot), \text{drop}(\cdot), \text{FC}.(\cdot)$ represent the Long Short-Term Memory (LSTM) (Hochreiter & Schmidhuber, 1997), Gaussian Error Linear Unit (GELU) activation (Hendrycks & Gimpel, 2016), dropout (Srivastava et al., 2014), and Fully Connected layer (FC) respectively.

This combination of random mask and trajectory modeling significantly contributes to the robustness and generalization of our student policy in deployment, conquering the challenge of partial observations and unseen real-world variations. (More Sim2Real techniques in Appendix E.4)

## 4 EXPERIMENTS

In this section, we conduct extensive experiments in both the simulator and the real world, covering various random backgrounds, objects, and positions to answer the following questions: (1) What is the scaling relationship between spatial generalization and data amounts in the grasping task? (Sec. 4.2) and (2) What about the efficacy of our method, especially its generalization ability to different backgrounds, objects, and object spatial positions in real-world applications? (Sec. 4.3)

### 4.1 EXPERIMENTAL SETUP

**Tasks.** To validate the generalization of Manibox, we test our approach under the grasping task. In the simulator, the articulated arm and end effector are required to pick up a green apple randomly placed on a table in a $41cm * 30cm * 28cm$ (length*width*height) spatial scale, which is referred to as "Full Space." Also, real-world counterparts are established with alternative objects, backgrounds, and settings to further demonstrate the generalization ability. In both scenarios, three RGB images taken by wrist cameras and the exterior camera are taken as observations.

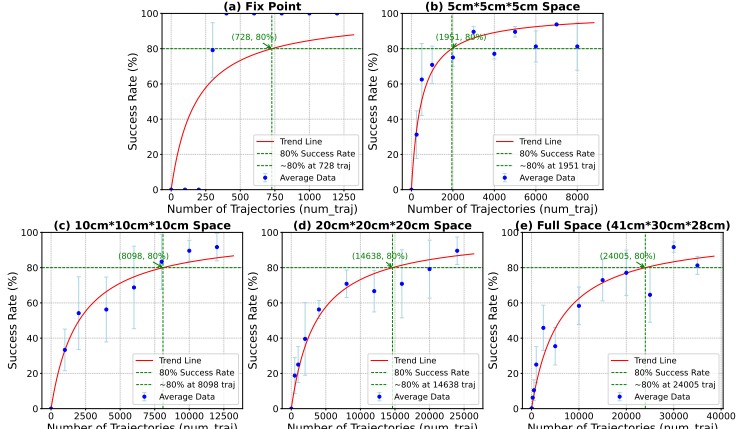

Figure 3: The scaling relationship between spatial generalization and data volume in the grasping task, measured under different spatial ranges. Data volume represents the number of trajectories used to train the student policy. The estimated 80% success point is marked, with blue points showing the average success rate across three seeds and error bars indicating standard deviation.

**Real Robot.** Mobile ALOHA (Fu et al., 2024) is a cost-effective and highly stable robot designed for both data collection and manipulation tasks. It owns four arms: two for teleoperation during data collection and two subordinate arms that perform manipulation tasks. Each subordinate arm is fitted with an RGB camera on its wrist, complemented by another RGB camera centrally mounted on the robot's body. Mobile ALOHA has demonstrated strong generalization capabilities across a diverse range of complex grasping tasks, making it an ideal choice for implementing our student policy.

## 4.2 SPATIAL SCALING LAW WITH DATA

To guide us in determining the data volume required for a certain level of spatial generalization, we investigate the relationship between policy performance and data volume. We conduct extensive experiments in the simulator to evaluate the success rate of policies trained on different data volumes. We select five distinct spatial ranges to explore this scaling behavior, including one fixed range and four dynamic ranges of 5, 10, 20 units, and full space. For each spatial range, we select more than 8 data volumes to train the student policy and evaluate its grasping success rate at random positions.

The results shown in Fig. 3 demonstrate that a scaling law holds in this context: For each spatial range, the pattern is as follows: when the data volume is zero, the policy inevitably fails. As the data volume increases, the success rate of the grasping policy improves, but the growth slows down. Eventually, the success rate surpasses 80% and tends to 100%.

Furthermore, we identify the data volume required to achieve an 80% success rate for each spatial range (as

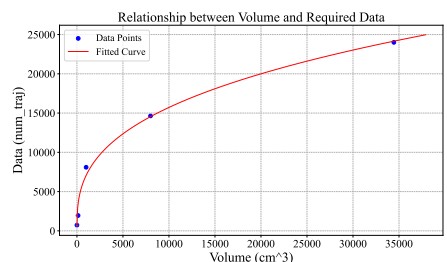

Figure 4: The relationship between spatial volume and data amounts needed to reach 80% grasping success rate. The fitted curve represents a power function $y = 640.32 \cdot x^{0.35}$.

shown in Fig. 3) and find that the volume of the spatial range and the required data follow a power rate relationship (as shown in Fig. 4). The positive correlation between data volume and spatial volume provides guidance for data-driven spatial generalization: The more data we have, the higher the robot's manipulation success rate. As the spatial range increases, more data is required, and the data volume needed grows with the space volume as a power rate relationship. In fact, for the same task, achieving high generalization success over a $41cm * 30cm * 28cm = 34440cm^3$ range may require at least 20,000 trajectory data from different spatial positions. Such large-scale data collection would be expensive in the real world, making simulator-generated data a more affordable and efficient option. This further emphasizes the significance of simulation data.

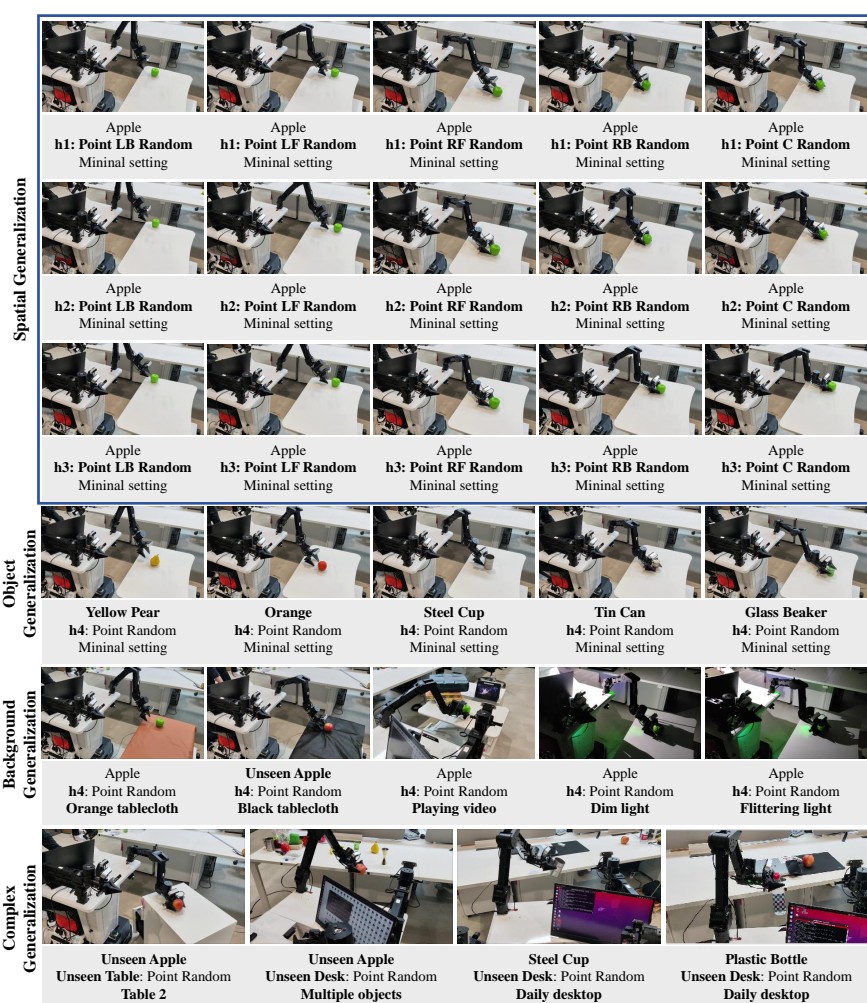

Figure 5: Demonstration of the generalization of our methods across different backgrounds, objects, and positions. The h1, h2, h3, h4 are randomly selected object heights in the real world, which are approximately 57cm, 65cm, 72cm, and 65cm. LB, LF, RF, RB, and C are abbreviations for Left-Back, Left-Front, Right-Front, Right-Back, and Center area respectively.

To further demonstrate the importance of other variables (e.g. time) in spatial intelligence and the robustness of policy generalization to visual uncertainty, we have further discussion on the spatial intelligence of our policy through detailed ablation experiments. Details are provided in Appendix B.

## 4.3 REAL-WORLD RESULTS

To validate the generalization ability of our method and policy in the real world, we compare the success rate of our policy with the vision-based ACT. In practice, we deploy models trained on simulation data onto a real robot and test their performance on grasping tasks across multiple arbitrarily chosen positions, eight objects(with some of them out-of-distribution), five backgrounds, and three complex settings as illustrated in Figure 5. More tasks, pictures, and videos of the experiment are available in the Appendix C and supplementary materials. More experiment details of these tests are in the Appendix D.3.

**Spatial generalization.** To justify the efficacy of our method, we assess the generalization capability of our student policy across four spatial ranges and a fixed point. For each experiment, we randomly place an apple, different from its counterpart in the simulator in both shape and material, on the table at arbitrarily chosen heights within the given range(as illustrated in the first three rows of Fig. 5). Alongside the difference in form and appearance, the apple presents considerable

unpredictability to the visual detection model due to factors including background, lighting, target object, etc. Consequently, the bounding box varies in size or even disappears at times, emphasizing the need for robustness in the policy. As shown in Tab. 1, our policy generalizes well to objects falling in different spatial volumes, achieving a success rate between 70% and 100%. In contrast, vision-based ACT, which trains the student policy with the vision input ACT (Zhao et al., 2023) on video-action simulation data, experiences a visible drop on success rate when spatial range scales up, despite its considerable success rate of 70% on a fixed point. The wanting performance of the baseline further demonstrates the significance and necessity of exploring spatial generalization. The results demonstrate that our method harbors strong generalization when addressing uncertainties arising from real-world objects, variations in spatial positions, and visual detection interferences.

**Object generalization.** In addition, to justify the object generalization ability of our method, we test our model across eight objects including original apple, unseen apple, yellow pear, orange, steel cup, tin can, glass beaker, and plastic bottle at arbitrarily selected positions. For each unseen object, feeding the corresponding name as a supplement into

Table 1: The real-world success rate of our method and vision-based ACT under varied spatial volumes. We select an approximate data amount based on the results in Fig. 3.

| SPATIAL RANGE | METHODS | SUCCESS RATE |
|---|---|---|
| Fix Point | Vision-Based ACT | 70% |
| | ManiBox (Ours) | 100% |
| 5cm*5cm*5cm | Vision-Based ACT | 0% |
| | ManiBox (Ours) | 70% |
| 10cm*10cm*10cm | Vision-Based ACT | 0% |
| | ManiBox (Ours) | 100% |
| 20cm*20cm*20cm | Vision-Based ACT | - |
| | ManiBox (Ours) | 80% |
| Full Space (41cm*30cm*28cm) | Vision-Based ACT | - |
| | Vision-Based ACT (100 real-world data) | 0% |
| | ManiBox (Ours) | 90% |

the open-vocabulary visual detection model is enough for the policy to grasp the target object. Tab. 4 shows detailed object size information for training and inference. It is worth noting that five of the selected objects are out-of-distribution in size (their bounding box sizes are unseen in the simulation data) in an attempt to further illustrate the object generalization ability of our method. This certain level of generalization observed, even when there is significant variation in the bounding box, aligns with our Lemma 2, which proves that the 3D position information of the object is complete in the dataset.

**Background Generalization.** Utilizing the open-vocab detection ability of YOLO-World, ManiBox can handle various backgrounds and vision disruptions without intentionally varied visual domain randomization. To verify this ability, we experiment with tables of three different shapes and tablecloths in two colors under the minimal setting of a table, an object, and a robot. In addition, we alter the lighting conditions by employing a single stationary point light source and a single flickering point light, as well as playing videos in the background as distractions. Furthermore, we construct two more complex scenarios, one representing a multi-object environment and the other replicating a typical daily workplace desktop. The results demonstrate the strong background generalization ability of our policy. Even though the visual detection model occasionally makes mistakes, the generalization of our policy compensates for those errors. This can be attributed to our policy's endeavor to integrate historical information and random masking to increase the visual model's tolerance for errors. Arguments for them can be found in Appendix B.

## 5 CONCLUSION

In this work, we present ManiBox, a novel bbox-guided manipulation method, using a simulation-based teacher-student framework to enhance the spatial generalization of grasping policies. To address the challenge of the generalization gap in vision-based models from simulation to real-world deployment, we propose a generalizable, state-based student policy that is innovatively equipped with bbox positions. This policy is trained on scalable simulation data generated by the teacher policy. Extensive experiments in both simulated and real-world environments demonstrate that ManiBox significantly outperforms existing methods in adapting to diverse spatial positions, object configurations, and background variations. Moreover, our theoretical and empirical analyses reveal that the generalization of spatial volume scales positively with the data volume, and the success rate follows Michaelis-Menten kinetics for specified spatial volumes. These findings pave the way for the development of robust spatial generalization across a wide range of embodied systems.

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

## A    PROOFS

### A.1    DETAILS AND PROOF OF THE LEMMA 1

We follow the setting in Crammer & Mansour (2012). First, assume the hypothesis set of policies to grasp a fixed single point is $\mathcal{H}$ and its VC dimension is $d$. When considering $T$ tasks each represented by a different grasping position, we take the $k$-shared task classifier $(\mathcal{H}_k, g)$ (Crammer & Mansour, 2012). Here $\mathcal{H}_k = \{h_1, ..., h_k\} \subseteq \mathcal{H}$ is a subset of $\mathcal{H}$ and $g$ maps each task $t \in \mathcal{T}$ to some $h_i \in \mathcal{H}_k$. Thus the $k$-shared task classifier hypothesis is defined as $\mathcal{F}_{\mathcal{H},k} = \{f_{\mathcal{H}_k,g} : |\mathcal{H}_k| = k, \mathcal{H}_k \subseteq \mathcal{H}, g : \mathcal{T} \to K\}$, here $f_{\mathcal{H}_k,g}(\cdot, t) = h_{g(t)}(\cdot)$. Then Crammer & Mansour (2012) has shown that the VC dimension of $\mathcal{F}_{\mathcal{H},k}$ is at most $\mathcal{O}(T \log k + kd \log(Tkd))$.

As we consider $\mathcal{H}$ as the set of neural networks, previous works have shown that the VC dimension of multi-layer neural networks is around the parameters of the neural networks (Maass, 1994; Sontag et al., 1998). Thus $d \gg T$ in our setting and $\mathcal{O}(T \log k + kd \log(Tkd)) \approx \mathcal{O}(kd \log(Tkd))$.

Finally, we consider the task number $T$. Assume that grasping two points of which the distance below $\epsilon$ is similar, we need around $b^3 / \frac{4}{3}\pi\epsilon^3$ points for covering the grasping cube with size $b \times b \times b$. Thus the VC dimension is at most

$$\mathcal{O}(kd \log(Tkd)) = \mathcal{O}(kd \log(\frac{b^3}{\frac{4}{3}\pi\epsilon^3} kd)) = \mathcal{O}(kd \log(\frac{3b^3 kd}{4\pi\epsilon^3})). \tag{5}$$

### A.2    PROOF OF THE LEMMA 2

*Proof.* Firstly, we convert normalized coordinates to pixel coordinates:

$$u_{i\min(\text{pixel})} = u_{i\min} \times W_i$$
$$v_{i\min(\text{pixel})} = v_{i\min} \times H_i$$
$$u_{i\max(\text{pixel})} = u_{i\max} \times W_i$$
$$v_{i\max(\text{pixel})} = v_{i\max} \times H_i$$

Then, we compute the center of the bounding box in pixel coordinates:

$$u_{i\text{center}} = \frac{u_{i\min(\text{pixel})} + u_{i\max(\text{pixel})}}{2}$$
$$v_{i\text{center}} = \frac{v_{i\min(\text{pixel})} + v_{i\max(\text{pixel})}}{2}$$

These points represent the projections of the sphere's center onto the image planes.

Secondly, for each camera, we back-project the center point into a 3D viewing ray in camera coordinates:

$$\mathbf{d}_i = K_i^{-1} \begin{bmatrix} u_{i\text{center}} \\ v_{i\text{center}} \\ 1 \end{bmatrix}$$

This gives the direction vector $\mathbf{d}_i$ in the camera coordinate system.

Therefore, we can express the sphere center in camera coordinates since the sphere's center lies somewhere along the viewing ray:

$$\mathbf{C}_{i,\text{cam}} = \lambda_i \mathbf{d}_i$$

where $\lambda_i > 0$ is an unknown scalar representing the distance along the ray from the camera center to the sphere's center.

Thirdly, we transform the sphere center from camera coordinates to world coordinates:

$$\mathbf{C} = R_i^\top \mathbf{C}_{i,\text{cam}} + \mathbf{t}_i = R_i^\top (\lambda_i \mathbf{d}_i) + \mathbf{t}_i$$

This must hold true for both cameras:

$$\mathbf{C} = R_1^\top (\lambda_1 \mathbf{d}_1) + \mathbf{t}_1 = R_2^\top (\lambda_2 \mathbf{d}_2) + \mathbf{t}_2$$

By equating the expressions for $\mathbf{C}$ we get:

$$R_1^\top (\lambda_1 \mathbf{d}_1) + \mathbf{t}_1 = R_2^\top (\lambda_2 \mathbf{d}_2) + \mathbf{t}_2$$

By rewriting we have:

$$R_1^\top \mathbf{d}_1 \lambda_1 - R_2^\top \mathbf{d}_2 \lambda_2 = \mathbf{t}_2 - \mathbf{t}_1$$

Let's denote:

$$\mathbf{a}_1 = R_1^\top \mathbf{d}_1$$
$$\mathbf{a}_2 = R_2^\top \mathbf{d}_2$$
$$\mathbf{b} = \mathbf{t}_2 - \mathbf{t}_1$$

So the equation becomes:

$$\mathbf{a}_1 \lambda_1 - \mathbf{a}_2 \lambda_2 = \mathbf{b}$$

Note that this is a system of three linear equations with two unknowns $\lambda_1$ and $\lambda_2$:

$$\begin{cases} a_{1x}\lambda_1 - a_{2x}\lambda_2 = b_x \\ a_{1y}\lambda_1 - a_{2y}\lambda_2 = b_y \\ a_{1z}\lambda_1 - a_{2z}\lambda_2 = b_z \end{cases}$$

Since we have more equations than unknowns (overdetermined system), we can solve for $\lambda_1$ and $\lambda_2$ using least squares estimation.

Note that this can be written in matrix form:

$$\begin{bmatrix} a_{1x} & -a_{2x} \\ a_{1y} & -a_{2y} \\ a_{1z} & -a_{2z} \end{bmatrix} \begin{bmatrix} \lambda_1 \\ \lambda_2 \end{bmatrix} = \begin{bmatrix} b_x \\ b_y \\ b_z \end{bmatrix}$$

Let $A$ be the $3 \times 2$ matrix of coefficients, $\boldsymbol{\lambda}$ the vector of unknowns, and $\mathbf{b}$ the vector of constants:

$$A = \begin{bmatrix} a_{1x} & -a_{2x} \\ a_{1y} & -a_{2y} \\ a_{1z} & -a_{2z} \end{bmatrix}, \quad \boldsymbol{\lambda} = \begin{bmatrix} \lambda_1 \\ \lambda_2 \end{bmatrix}, \quad \mathbf{b} = \begin{bmatrix} b_x \\ b_y \\ b_z \end{bmatrix}$$

We can compute the least squares solution:

$$\boldsymbol{\lambda} = (A^\top A)^{-1} A^\top \mathbf{b}$$

Once $\lambda_1$ and $\lambda_2$ are found, **C can be calculated using either camera's equation**:

$$\mathbf{C} = R_1^\top (\lambda_1 \mathbf{d}_1) + \mathbf{t}_1$$

Alternatively, we compute both and average them for robustness:

$$\mathbf{C} = \frac{1}{2} \left( R_1^\top (\lambda_1 \mathbf{d}_1) + \mathbf{t}_1 + R_2^\top (\lambda_2 \mathbf{d}_2) + \mathbf{t}_2 \right)$$

Finally, since $\lambda_i$ and $\lambda_2$ are found, we can compute the distance $Z_i$ from the camera along the viewing direction $\mathbf{d}_i$:

$$\mathbf{C}_{i,\text{cam}} = \lambda_i \mathbf{d}_i = Z_i \cdot \hat{\mathbf{d}}_i$$

where $\hat{\mathbf{d}}_i$ is normalized by:

$$\hat{\mathbf{d}}_i = \frac{\mathbf{d}_i}{\|\mathbf{d}_i\|}$$

The projected radius of the sphere $s_i$ (in pixels) is half the size of the bounding box width or height (assuming the sphere projects to a circle):

$$s_i = \frac{w_i + h_i}{4}$$

where

$$w_i = u_{i\max(\text{pixel})} - u_{i\min(\text{pixel})}$$
$$h_i = v_{i\max(\text{pixel})} - v_{i\min(\text{pixel})}$$

Consider similar triangles in the imaging geometry:

$$\frac{s_i}{f_i} = \frac{r}{Z_i}$$

**We can compute the radius of the sphere**:

$$r = \frac{s_i \cdot Z_i}{f_i}$$

Additionally, the solution is unique provided that:

- **The Cameras Are Not Aligned:** The direction vectors $\mathbf{a}_1$ and $\mathbf{a}_2$ are not colinear.

- **Non-Degenerate Configuration:** The cameras have different positions or orientations, ensuring the lines of sight intersect at a single point.

Under these conditions, the least squares solution yields a unique $\lambda$, and thus unique $\mathbf{C}$ and $r$.

$\square$

## B   ABLATION STUDY

To demonstrate the essential role of 3D spatial and 1D temporal information for spatial intelligence, we perform an additional experiment focusing on **temporal variability**, in addition to the spatial generalization tests in Sec. 4. This experiment includes an ablation study comparing models trained on a few selected key timesteps with those trained on all timesteps in a full trajectory. To further illustrate the robust inference capabilities achieved by the enhanced **policy generalization**, we test the policy's ability to handle significant uncertainty in the output of the visual model. During inference, we apply random masks and noise to the visual model's bounding box outputs, allowing us to evaluate the policy's generalization ability under such conditions.

### B.1   TRAINING WITH ONLY KEY STEPS.

To assess the impact of temporal information on policy effectiveness, we conduct an experiment where the model is trained using only key steps from the trajectory. The specific key steps selected were [0, 18, 20, 22, 70], chosen to represent pivotal moments in the trajectory for the grasping task: 0 corresponds to the initial state before the gripper starts approaching the apple, 70 represents the final state when the robotic arm retracts, and steps 18 to 22 denote intermediate states where the gripper makes contact with the apple and closes around it.

Table 2: Success Rates for Training with Only Key Steps

| Experiment | Real Robot Success Rate | Simulation Success Rate |
|---|---|---|
| Training with Full Trajectory | 90% | 81.25% |
| Training with Key Steps | 0% | 4.17% |

The results in Tab. 2 indicate a stark difference in performance between training with the full trajectory and training with only key steps. Training with the full trajectory resulted in a success rate of 90% on the real robot and 81.25% in simulation. In contrast, training with only key steps leads to complete failure on the real robot (0% success rate) and a significantly reduced success rate of 4.17% in simulation. This outcome highlights that relying solely on key steps fails to provide the temporal continuity necessary for effective spatial generalization. The policy's inability to capture sufficient context from isolated states results in poor action generation and overall task failure.

### B.2   ROBUSTNESS AGAINST DETECTION FAILURE DURING INFERENCE.

During the inference phase, visual models such as YOLO-World may not always successfully identify objects, leading to significant uncertainty in visual input.

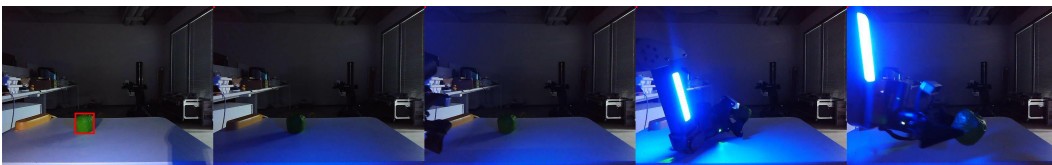

Figure 6: A case of detection failure. The task is to catch apples to a specified location in a normal environment.

Figure 7: A case of detection failure. The task is to catch apples to a specified location in a flittering light environment.

**Our policy still works well in situations where the visual model fails to detect.** For example, visual models may fail to detect target objects when the robotic arm approaches the target, under dim lighting conditions, or when there is partial occlusion of the target by the background (see Figures 6 and 7). Especially, when using flickering and swaying point light sources in dimly lit environments, the detection by visual models is particularly unstable. Such failures may occur at any timestep during manipulation and may persist for several timesteps. Moreover, these failures are unstable and unpredictable. For the same object placement, the occurrence of detection failures during manipulation does not consistently happen or not happen at the same timestep.

Despite all those failures and instabilities in detecting bounding boxes, our model can still successfully and accurately accomplish the manipulation tasks.

To test our policy's robustness against this visual uncertainty, we introduce masks that randomly obscured bounding box input at varying ratios (0.0 to 1.0) during inference.

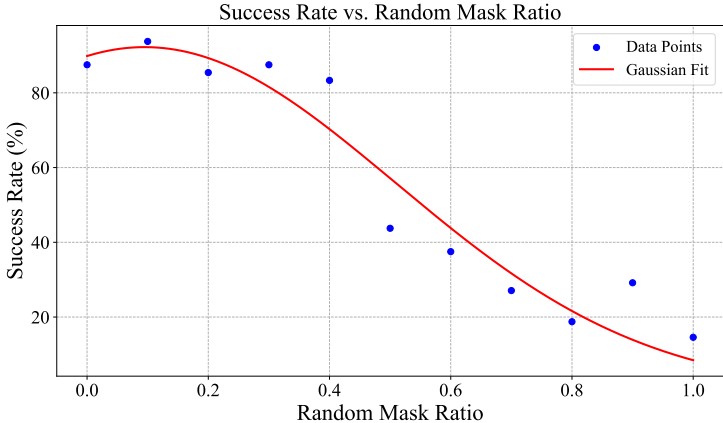

Figure 8: The relationship between random mask ratio and success rate. The fitted curve represents a Gaussian function of $y = 92.23 \exp\left(-\frac{(x-0.09)^2}{2 \cdot 0.41^2}\right)$.

As shown in Fig. 8, the results reveal that the policy maintains a high success rate and generalization capability when the mask ratio is below 0.4. As the mask ratio exceeds 0.4, the success rate significantly declines, leading to consistent policy failures. Coincidentally, the random mask ratio we add in the student policy training phase is 0.3. This trend reveals that our approach focusing on policy generalizability can resist visual uncertainty up to 40% of the masking ratio. That is, even if 40%

of the visual information is lost, the policy still has good spatial generalizability. We fitted the data points with a Gaussian curve to represent this trend, defined by the equation:

$$y(x) = a \cdot e^{-\frac{(x-b)^2}{2c^2}} \tag{6}$$

where $a$, $b$, and $c$ are parameters that define the height, center, and width of the curve, respectively. This curve demonstrates that the policy is resilient up to moderate levels of visual masking (mask ratios up to 0.3), but performance degrades significantly as the masking ratio increases beyond 0.4.

This experiment validates that our policy can withstand moderate visual uncertainties during inference, maintaining robust performance. This shows that our focus on policy generalization can easily cope with situations where the visual model has large uncertainty, leading to more robust manipulation. However, the steep decline in success rate for higher mask ratios highlights the importance of reliable visual input for effective spatial reasoning and task execution.

As to whether the random mask in the training phase gives some increase in the success rate of our policy, please refer to Sec. B.3.

### B.3 Random Mask during Student Policy Training Affects Performance.

The uncertainty of visual detection in the real world is mentioned in the previous subsection, in order to demonstrate that random masking during training does favor the performance of the student policy, Specifically, we compare the success rates of policies trained with and without a random mask ratio of 0.3 across five different spatial ranges. During training, we randomly mask out 0 to 30% of the bounding box information in each trajectory. This encourages the policy to adapt to scenarios where bounding box detection is incomplete or inaccurate. Table 3 demonstrated that in all range of spaces, the policy trained with random masks shows outstanding performance compared with one without masks. In addition, the policy shows better robustness in larger spatial ranges, where the variability and uncertainty of detection are more pronounced, highlighting the effectiveness of random masking in improving spatial adaptability and generalization.

Table 3: Comparison of Success Rates With and Without Random Mask

| Spatial Range | Random Mask Ratio = 0.3 | Without Random Mask |
|---|---|---|
| Fix Point | **100.0%** | **100.0%** |
| 5 cm × 5 cm × 5 cm | **83.33%** | 79.17% |
| 10 cm × 10 cm × 10 cm | **97.92%** | 64.58% |
| 20 cm × 20 cm × 20 cm | **87.5%** | 70.83% |
| Full Space (41 cm × 30 cm × 28 cm) | **87.5%** | 68.75% |

### B.4 Random Noise in Bounding Boxes during Inference.

To further examine the policy's robustness against visual uncertainty, we introduce random noise to the bounding boxes during inference. We tested noise ratios at increments of 0.01, 0.02, 0.05, 0.1, 0.2, and 0.5. Fig. 9 illustrates the success rate as a function of noise ratio, where a monotonic decay fit highlights the policy's performance trend. The results show that the policy maintains a high success rate at low noise ratios but experienced a significant decrease as noise levels increased. Since our normalized bounding box value ranges from 0 to 1, the policy still achieves a high success rate with a visual detection error of 0.05, demonstrating the robustness of the policy generalization to cope with visual uncertainty and visual sim2real gap.

Our experiments highlight that achieving spatial intelligence requires both continuous temporal information and sufficient spatial data. The experiments in our main text demonstrate that spatial generalizability requires sufficient spatially varying data. The first ablation experiment shows that continuous temporal context is crucial for effective policy performance, while the second and third ablation experiments demonstrate that policy generalization can cope with notable visual uncertainty.

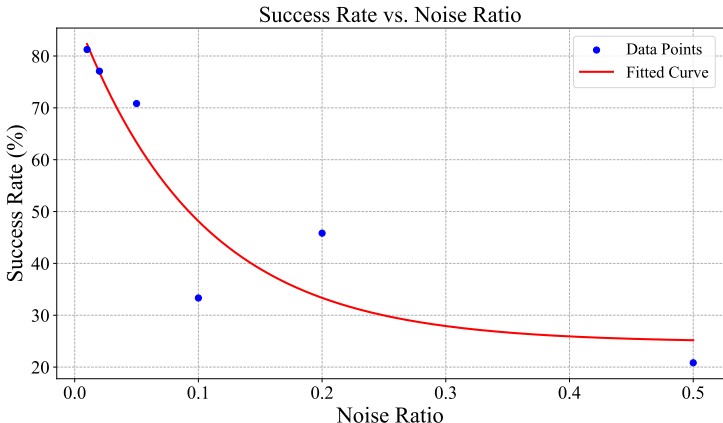

Figure 9: The relationship between random mask ratio and success rate. The fitted curve represents a exponential function of $y = 63.62 \exp(10.01x) + 24.75$.

## C  MORE REAL-WORLD EXPERIMENTS

To demonstrate the extensibility of our approach, we supplement four more real-world experiments: **Pour Water** (Figure 10), showcasing our approach's extensibility to multi-object manipulation tasks; **Grab the handle of the cup** (Figure 11), illustrating our approach's extensibility to the grasping of detailed parts of irregular objects; **Mid-Air** (Figure 12); and **Cluttered Table** at varying heights (Figure 13 and Figure 14).

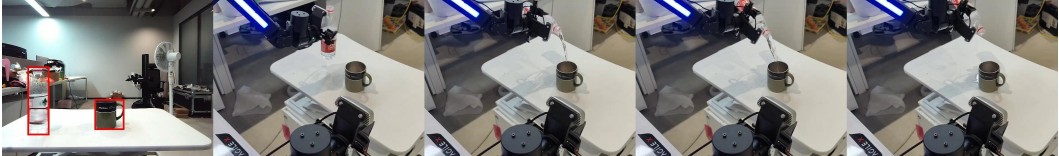

Figure 10: **Task: Pour Water.** The policy grasps the left water bottle and pours water into the right cup, demonstrating the extensibility of our approach to complex tasks. By modifying the teacher policy in the simulator, this method can be extended to **more sophisticated tasks with more objects**.

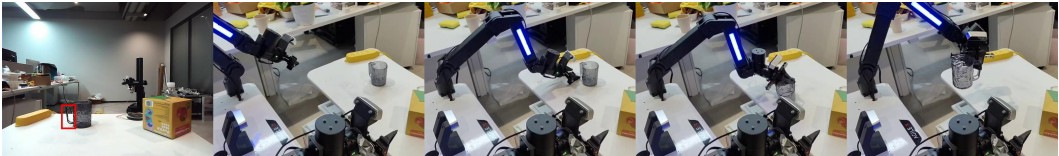

Figure 11: **Task: Grab the handle of the cup**. More fine-grained detection can be achieved using models such as Liu et al. (2023), so that our policy can catch irregular objects, such as the handle of a cup. The first figure shows the recognition of the cup handle from the first-person view and the other figures show the task completion process from the third-person view.

## D  EXPERIMENTAL DETAILS

We test the success rate of student policy in both simulator and real world.

### D.1  EVALUATION IN SIMULATION

We run large-scale experiments in the simulator to test the spatial generalization curve for each range and determine the minimum data amounts to achieve an optimal success rate. To align with real robot

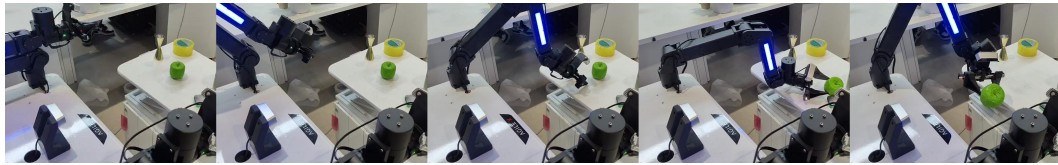

Figure 12: **Task: Mid-Air.** Grasp the bottle lifted by hand and suspended in mid-air, then move it to the target position.

Figure 13: **Task: Cluttered Table.** Grasp the apple from a cluttered table with a height of 60 cm.

inference, we set the robot's joints directly to target qpos instead of using environment stepping. We mark the trajectory as success if there is a step both reaching reward $> 0.6$ and quat reward $> 1.4$. For each policy, we test 48 trajectories and record the final success rate. The experiment consists of three rounds of tests with different seeds, each with 16 trajectories. In each round, the target random points are sampled from a uniform distribution.

## D.2 EXPERIMENT RESULT ANALYSIS

We discuss here the analysis of the results of spatial scaling law.

$$\text{data\_volume} = 640.32 \cdot \text{spatial\_volume}^{0.35} \tag{7}$$

By fitting the resulting data points, we observe that as the training data increases, the success rate of the policies quickly rises and eventually converges to an upper bound, which aligns well with the form of the Michaelis-Menten (Michaelis et al., 1913) equation:

$$\text{success\_rate} = \frac{V_{\max} \times \text{num\_traj}}{K_m + \text{num\_traj}} \tag{8}$$

where $V_{\max}$ represents the maximum achievable success rate, and $K_m$ is the number of trajectories at which the success rate is half of $V_{\max}$. This equation captures the saturation-type growth observed in the policy performance.

Across different settings, the amount of data required to achieve an 80% success rate shows a power function as the data increases:

For the functional relationship between the amount of data required for spatial generalization and spatial volume, we explored two types of fits: a power law and a logarithmic function. The fitting results of power law is given by $y = a \cdot x^b = 640.32 x^{0.35}$, while the logarithmic fit is $y = a \cdot \ln(b \cdot x) + c = 2095.51 \cdot \ln(0.3 \cdot x) - 642.93$. We found that the logarithmic fit becomes unreasonable as the spatial volume $x$ approaches 0, where the required data, $y = a \cdot \ln(b \cdot x) + c = 2095.51 \cdot \ln(0.3 \cdot x) - 642.93$, tends to negative infinity, which is unrealistic since the amount of data cannot be negative. Furthermore, the fit results in a negative value at $x = 3.3$, where $bx = 1$ and $\log(bx) = 0$, leading to a required data amount of $a \log(bx) + c = -642.93$. In the long run (for larger spatial volumes), the power law fit better describes the relationship between the data required for spatial generalization and spatial volume. Fig. 15 shows the selected power law relationship, and Fig. 16 displays the logarithmic curve.

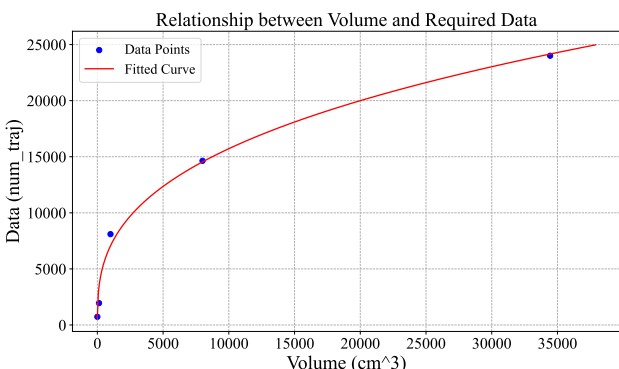

Figure 14: **Task: Cluttered Table.** Grasp the apple from a cluttered table with a height of 74.5 cm.

Figure 15: Demonstration of the power-rate relationship between spatial volume and data amounts needed to reach 80% success rate in the grasping task.

## D.3 EVALUATION IN REAL WORLD

### D.3.1 REAL-WORLD SUCCESS RATE

To verify our experiment results from the simulator, we evaluate our policy in the real robot. For each range, we choose the first optimal policy that achieves 80% success rate in simulator and evaluate with the real robot for 10 randomly chosen points.

**Random Placement of Objects in Real-world Evaluation.**   To be more specific, we divide the maximum range, namely the full space ( $41cm * 30cm * 28cm$ ), into three equal parts in height, two equal parts in width, and three equal parts in length, resulting in a total of $3 * 2 * 3 = 18$ equally sized spatial cuboid ranges at different positions. For each tested spatial range, ten different points are uniformly selected to test the robot's manipulation success rate.

### D.3.2 REAL-WORLD EXPERIMENT SETTINGS

To verify our methods, we evaluate our policy in the real robot. We evaluate the student policy which is trained on simulation data of full space ( $41cm * 30cm * 28cm$ ) with the real robot.

**Spatial Generalization.**   As illustrated in Fig. 5, we divide the spatial range horizontally into Left (L), Right (R), and Center (C), and vertically into Front (F) and Back (B), and randomly select points within these sections on different heights to test our policy[2]. All these points are arbitrarily selected, which means that the target object has an equal chance of appearing at the center or the edge of the table, far from or close to the robot, in positions that require the robotic arm to reach upwards or downwards. We hope this approach reflects the spatial generalization of our policy comprehensively and honestly.

In addition to the verification of spatial generalization, we also test the ability of Object generalization and background generalization. The details of the experiment setting can be found in Tab. 4.

**Object Generalization and Background Generalization.**   Furthermore, we have chosen objects of various shapes, sizes, and materials as target objects for testing, all of which were previously

---

[2]The table is an elevated table and can therefore be adjusted to any height.

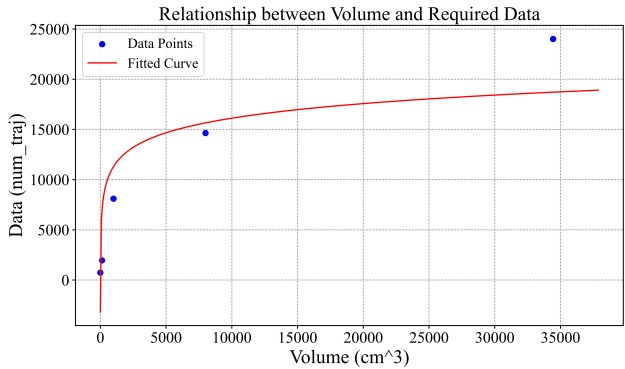

Figure 16: Demonstration of the logarithmic relationship between spatial volume and data amounts needed to reach 80% success rate in the grasping task.

unseen by our policy. To be more specific, during training, we only encountered apples with sizes of approximately $(6.9cm - 7.4cm, 6.9cm - 7.4cm, 6.9cm - 7.4cm)$ in the XYZ direction, while the test target objects span a size range of $(5.1cm - 7.0cm, 5.1cm - 9.7cm, 6.0cm - 22.7cm)$ in the XYZ direction. Additionally, although apples are roughly spherical, the target objects also include cylindrical and pyriform shapes. Moreover, the target objects encompass different materials such as glass and metal. All these objects serve to demonstrate our policy's capability to generalize to varying shapes, sizes, and materials.

Table 4: Real-World Experiment Settings. Note that we use only the bounding boxes of the objects for both student policy training and inference. Bolded object sizes are never seen in training.

| Generalization Type | Object Size (cm) of XYZ | Setting Description |
|---|---|---|
| **Object in Simulation Data** | | |
| Apple (Domain Randomization) | (6.9-7.4, 6.9-7.4, 6.9-7.4) | Distribute uniformly in a configurable spatial volume |
| **Object Generalization** | | |
| Yellow Pear | **(6.5, 5.5, 10.7)** | h4: Point Random, Minimal setting |
| Orange | **(7.0, 6.7, 6.5)** | h4: Point Random, Minimal setting |
| Steel Cup | **(6.5, 9.7, 10.8)** | h4: Point Random, Minimal setting |
| Tin Can | **(5.1,5.1,7.2)** | h4: Point Random, Minimal setting |
| Glass Beaker | **(5.1, 5.1, 6.0)** | h4: Point Random, Minimal setting |
| **Background Generalization** | | |
| Apple | (7.2, 7.4, 6.9) | h4: Point Random, Orange tablecloth |
| Unseen Apple | **(7.0, 7.5, 7.0)** | h4: Point Random, Black tablecloth |
| Apple | (7.2, 7.4, 6.9) | h4: Point Random, Playing video |
| Apple | (7.2, 7.4, 6.9) | h4: Point Random, Dim light |
| Apple | (7.2, 7.4, 6.9) | h4: Point Random, Flittering light |
| **Complex Generalization** | | |
| Unseen Apple, Unseen Table | **(7.0, 7.5, 7.0)** | Point Random, Table 2 |
| Unseen Apple, Unseen Desk | **(7.0, 7.5, 7.0)** | Point Random, Multiple objects |
| Steel Cup, Unseen Desk | **(6.5, 9.7, 10.8)** | Point Random, Daily desktop |
| Plastic Bottle, Unseen Desk | **(6.4, 6.4, 22.27)** | Point Random, Daily desktop |

The three camera images, as well as the entire trajectory of grasping, are schematized below in Figure 17:

# E    TEACHER-STUDENT TRAINING DETAILS

For ease of understanding, Alg. 1 is the pseudo algorithm for the ManiBox method.

---

**Algorithm 1** Training and Deployment Algorithm of ManiBox

---

**Require:** Simulator environment $Sim$, Real robot $Robot$
**Require:** Proximal Policy Optimization (PPO) for training teacher policy
**Require:** Bounding box extractor $BBoxExtractor$
**Require:** Maximum dataset size $N_{max}$, reward function $R$, domain randomization module $DR$

1: **Stage 1: Teacher Policy Training**
2: Initialize teacher policy $\pi_\beta$ with random weights
3: Set $Sim$ environment with reward function $R$ and domain randomization module $DR$
4: **while** not converged **do**
5:      Collect trajectory data $\mathcal{D}_T$ in $Sim$ using $\pi_\beta$
6:      Update $\pi_\beta$ using PPO with $\mathcal{D}_T$
7: **end while**

8: **Stage 2: Simulation Data Generation**
9: Initialize dataset $\mathcal{D}_S = \emptyset$
10: **while** $|\mathcal{D}_S| < N_{max}$ **do**
11:      Reset $Sim$ environment
12:      Apply domain randomization $DR$ to environment
13:      Generate trajectory $\tau = \{(o_t^{rob}, a_t, r_t)\}_{t=1}^T$ using $\pi_\beta$
14:      Extract bounding boxes $\{o_t^{vis}\}_{t=1}^T$ using $BBoxExtractor$
15:      Store $\{(o_t^{vis}, o_t^{rob}, a_t)\}_{t=1}^T$ in $\mathcal{D}_S$
16: **end while**

17: **Stage 3: Student Policy Training**
18: Initialize student policy $\pi_\theta$ with random weights
19: **while** not converged **do**
20:      Sample trajectory $\tau = \{(o_t^{vis}, o_t^{rob}, a_t)\}_{t=1}^T$ from $\mathcal{D}_S$
21:      Randomly mask some $o_t^{vis}$ to simulate detection failures
22:      Update $\pi_\theta(o_{\leq t}^{vis}, o_{\leq t}^{rob})$ using the masked $\tau$
23: **end while**

24: **Stage 4: Real-World Deployment**
25: $t \leftarrow 0$
26: **while** not terminated **do**
27:      Extract real-world bounding boxes $o_t^{vis}$ using $BBoxExtractor$
28:      Compute action $a_t \sim \pi_\theta(o_{\leq t}^{vis}, o_{\leq t}^{rob})$
29:      Execute action $a_t$ on $Robot$
30:      $t \leftarrow t + 1$
31: **end while**

---

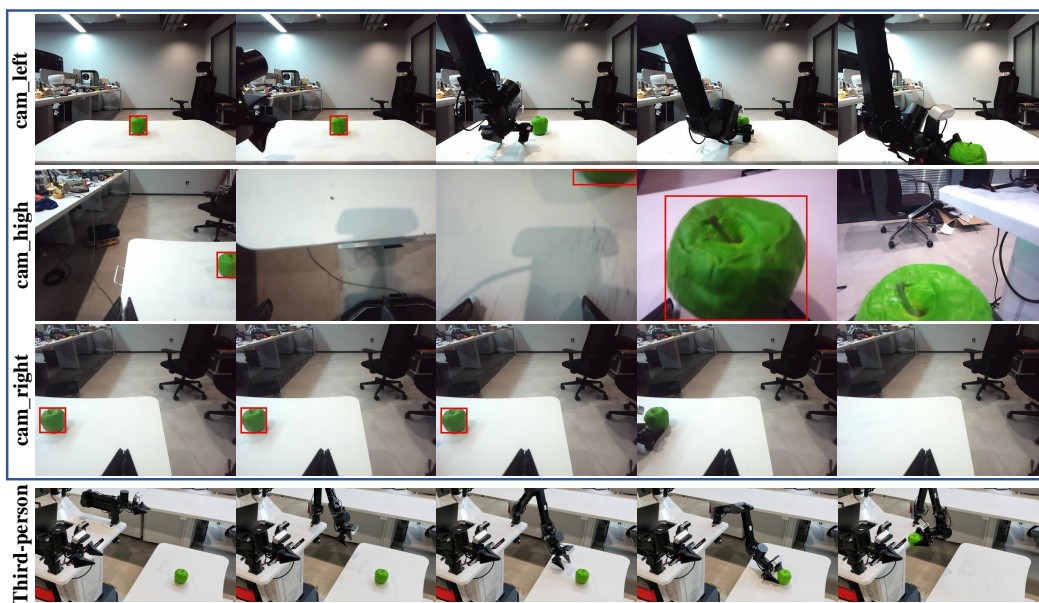

Figure 17: Pictures of the entire trajectory of grasping, including perspective and three camera views.

## E.1 TEACHER POLICY DETAILS

We train our teacher policy using PPO with 8,192 parallel environments in Isaac Lab (Mittal et al., 2023). With a carefully designed reward function and applied PPO techniques, the training is completed in around 40 minutes, reaching approximately 2,000 training iterations. The task involves training a robot to grasp an apple randomly placed on a liftable table and transport it to a designated target position (Figure 18).

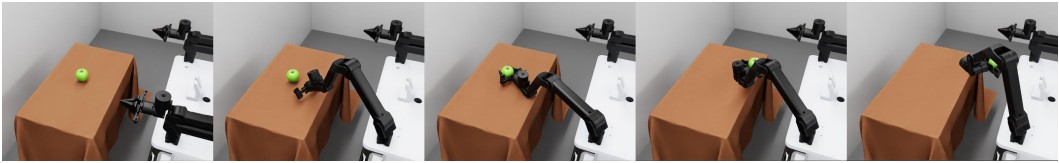

Figure 18: The simulation environment consists of a room with a table, an apple, and a robot. The task is for the robot to grasp the apple and place it in a specified location.

The advantages of using a teacher policy instead of direct visual RL training can be summarized in three key aspects. First, the teacher policy relies solely on robot and object information as input, significantly reducing the dimensionality of the state space. Second, the incorporation of privileged information and tailored reward functions for critical steps offers more focused guidance during training, thereby narrowing the exploration space. Lastly, visual RL suffers from limited parallelizability, which results in lower overall training efficiency.

### E.1.1 SIMULATION SCENARIO SETUP

The environment is built based on the lift task in Isaac Lab. During training, the scene consists of a table, an apple, and a robot. The robot is initialized at a fixed starting position at the beginning of each episode. The apple's size, as well as its x,y position, are uniformly randomized across environments, while the height of the table is also uniformly randomized. The target position, however, remains fixed throughout all training episodes. Key steps captured from three camera perspectives

are shown in Figures 19, 20, and 21. To support future extensions to more complex tasks, such as obstacle avoidance during grasping, we represent the robot's behavior with a 30Hz trajectory. Each environment in the simulator is evenly spaced to prevent interference between different training instances. Further details on the randomized ranges can be found in Table 5.

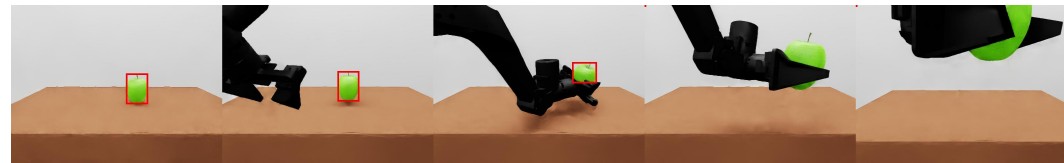

Figure 19: Key steps of grasping in simulation in cam_high view.

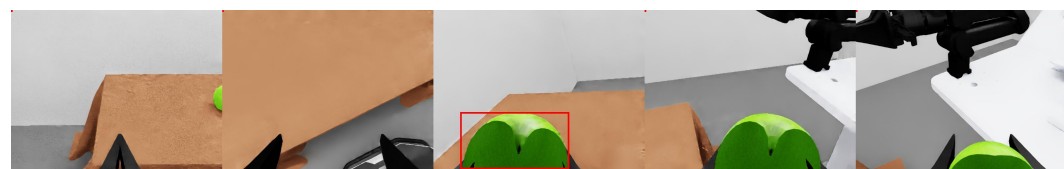

Figure 20: Key steps of grasping in simulation in cam_left_wrist view.

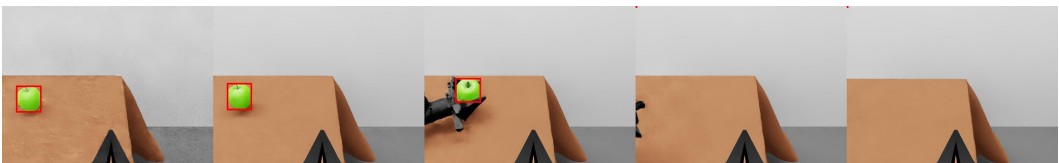

Figure 21: Key steps of grasping in simulation in cam_right_wrist view.

Table 5: Environment Randomization Parameters

| Spatial Range | Table Height (z) | Apple X Position | Apple Y Position |
|---|---|---|---|
| Fix Point | $(-0.00, 0.00)$ | $(-0.0, 0.0)$ | $(-0.0, 0.0)$ |
| $5\,\text{cm} \times 5\,\text{cm} \times 5\,\text{cm}$ | $(-0.025, 0.025)$ | $(-0.025, 0.025)$ | $(-0.025, 0.025)$ |
| $10\,\text{cm} \times 10\,\text{cm} \times 10\,\text{cm}$ | $(-0.07, 0.03)$ | $(-0.05, 0.05)$ | $(-0.05, 0.05)$ |
| $20\,\text{cm} \times 20\,\text{cm} \times 20\,\text{cm}$ | $(-0.1, 0.1)$ | $(-0.1, 0.1)$ | $(-0.05, 0.15)$ |
| Full Space ($41\,\text{cm} \times 30\,\text{cm} \times 28\,\text{cm}$) | $(-0.13, 0.15)$ | $(-0.22, 0.08)$ | $(-0.05, 0.36)$ |
| **Apple 3D Scale Range** | $((0.77, 0.82), (0.77, 0.82), (0.77, 0.82))$ | | |

### E.1.2 DETAILS OF OBSERVATION AND ACTION

We provide detailed information about the observations and actions used in training the teacher policy for the multi-environment object-grasping manipulation task:

- **Observations:**
    - Contact forces (24-dimensional)
    - Joint position (8-dimensional)
    - Joint velocity (8-dimensional)
    - Object position (3-dimensional)
    - Target location (7-dimensional)

– Last action (8-dimensional)

- **Actions:**

  – Joint Position (8-dimensional) (We choose position control. The action is the expected joint position for the next timestep.)

### E.1.3 IMPLEMENTATION OF PPO

Our PPO implementation is built on rsl_rl, and leverages several techniques to improve stability and performance. Key components and their settings are summarized in Table 6.

Table 6: PPO Parameter Settings

| Techniques | Parameter | Value |
|---|---|---|
| Clipped Surrogate Objective | clip_param | 0.2 |
| Value Function Clipping | use_clipped_value_loss | True |
| Value Loss Coefficient | value_loss_coef | 1.0 |
| Adaptive KL Penalty | desired_kl | 0.01 |
| Entropy Regularization | entropy_coef | 0.01 |
| Generalized Advantage Estimation (GAE) | gamma | 0.99 |
|  | lam | 0.95 |
| Gradient Clipping | max_grad_norm | 1.0 |
| Learning Rate | learning_rate | 0.001 |
| Mini-Batch | num_mini_batches | 4 |
| Learning Epochs | num_learning_epochs | 5 |
| Schedule | schedule | adaptive |
| Empirical Normalization | empirical_normalization | True |

### E.1.4 REWARD FUNCTION DESIGN

To ensure the policy achieves the desired grasping behavior on our robot, we design a reward function that reflects the task's objectives. As grasping is complicated, it can be divided into a multi-stage reward function: first reaching the object, then closing the jaws, then picking up the object, and finally retracting the robotic arm to a specified position. Definitions of different part of our reward function for the specific object-grasp task are stated as follows:

- **Fingers Open**

$$R_{\text{fingers\_open}} = \tanh\left(\frac{\|\mathbf{p}_{\text{object}} - \mathbf{p}_{\text{ee}}\|_2}{\sigma}\right) \cdot \sum_{i=1}^{2} \text{finger\_pos}[i]$$

where $\mathbf{p}_{\text{object}}$ represents the position of the object in world coordinates. $\mathbf{p}_{\text{ee}}$ represents the position of the end-effector in world coordinates. finger_pos represents the joint positions of the robot's fingers. $\sigma = 0.1$ is a scaling factor for distance normalization.

- **Reaching Object**

$$R_{\text{reaching\_object}} = \left(1 - \tanh\left(\frac{\|\mathbf{p}_{\text{object}} - \mathbf{p}_{\text{ee}}\|_2}{\sigma}\right)\right) + \left(1 - \tanh\left(\frac{\|\mathbf{p}_{\text{object}} - \mathbf{p}_{\text{ee}}\|_2}{\sigma/4}\right)\right)$$

where $\mathbf{p}_{\text{object}}$ represents the position of the object in world coordinates. $\mathbf{p}_{\text{ee}}$ represents the position of the end-effector in world coordinates. $\sigma = 0.2$ is a scaling factor for distance normalization.

- **Object Goal Tracking**

$$R_{\text{goal\_tracking}} = \mathbb{1}(\mathbf{p}_{\text{object},z} > z_{\text{init}} + h_{\text{min}}) \cdot \left(1 - \tanh\left(\frac{\|\mathbf{p}_{\text{goal}} - \mathbf{p}_{\text{object}}\|_2}{\sigma}\right)\right)$$

where $\mathbf{p}_{\text{goal}}$ represents the target position of the object in world coordinates. $\mathbf{p}_{\text{object}}$ represents the position of the object in world coordinates. $\mathbf{p}_{\text{object},z}$ represents the $z$-coordinate (height) of the object in world coordinates. $z_{\text{init}}$ is the initial height of the object.

$h_{\min} = 0.02$ is the minimal height above the initial height required to achieve the goal. $\sigma = 0.3$ is a scaling factor for distance normalization.

- **Object Goal Tracking (Fine-Grained)**

$$R_{\text{goal\_tracking\_fine}} = \mathbb{1}(\mathbf{p}_{\text{object},z} > z_{\text{init}} + h_{\min}) \cdot \left( 1 - \tanh\left( \frac{\|\mathbf{p}_{\text{goal}} - \mathbf{p}_{\text{object}}\|_2}{\sigma_{\text{fine}}} \right) \right)$$

where $\mathbf{p}_{\text{goal}}$ represents the target position of the object in world coordinates. $\mathbf{p}_{\text{object}}$ represents the position of the object in world coordinates. $z_{\text{init}}$ is the initial height of the object. $h_{\min} = 0.02$ is the minimal height above the initial height required to achieve the goal. $\sigma_{\text{fine}} = 0.05$ provides finer scaling for precise tracking.

- **Action Rate Penalty**

$$R_{\text{action\_rate}} = -\|\mathbf{a}_t - \mathbf{a}_{t-1}\|_2^2$$

where $\mathbf{a}_t$ represents the action at the current time step. $\mathbf{a}_{t-1}$ represents the action at the previous time step.

- **Joint Velocity Penalty**

$$R_{\text{joint\_vel}} = - \sum_{i \in \text{joint\_ids}} \dot{q}_i^2$$

where $\dot{q}_i$ represents the joint velocity of the $i$-th joint. joint_ids is the set of joint indices to penalize.

- **Contact Forces**

$$R_{\text{contact\_forces}} = \mathbb{1}\left( \frac{\text{finger\_dist}}{4} < \text{proj\_len} < \frac{\text{finger\_dist}}{1.25} + 0.2 \right) \cdot \left( 1.2 - \tanh\left( \frac{\text{object\_dist}}{0.2} \right) \right)$$

$$\cdot \mathbb{1}\left( \|\mathbf{p}_{\text{object}} - \mathbf{p}_{\text{top}}\| < 0.1 \right) \cdot \left( 1 - \tanh\left( \frac{\|\mathbf{org\_vec} - \mathbf{y}\|}{0.3} \right) \right)$$

where $\mathbf{p}_{\text{top}}, \mathbf{p}_{\text{bottom}}, \mathbf{p}_{\text{left}}, \mathbf{p}_{\text{right}}$ are the positions of the robot's end-effector fingers (top, bottom, left, right) in world coordinates. $\mathbf{p}_{\text{object}}$ is the position of the object in world coordinates. $\mathbf{org\_vec} = \mathbf{p}_{\text{left}} - \mathbf{p}_{\text{right}}$ is the vector between the left and right fingers, rotated into the object's local frame. $\mathbf{y} = [0, 1, 0]$ is the unit vector along the y-axis in the object's local frame. $\text{finger\_dist} = \|\mathbf{p}_{\text{left}} - \mathbf{p}_{\text{right}}\|_2$ is the distance between the left and right fingers. $\text{proj\_len} = \frac{(\mathbf{p}_{\text{object}} - \mathbf{p}_{\text{left}}) \cdot (\mathbf{p}_{\text{right}} - \mathbf{p}_{\text{left}})}{\text{finger\_dist}}$ is the projection of the object's position onto the vector between the left and right fingers. $\text{object\_dist} = \frac{\|\mathbf{p}_{\text{object}} - \mathbf{p}_{\text{bottom}} \times (\mathbf{p}_{\text{top}} - \mathbf{p}_{\text{bottom}})\|}{0.11}$ is the perpendicular distance from the object to the line formed by the bottom and top fingers.

- **Close Fingers**

$$R_{\text{close\_fingers}} = \mathbb{1}(\text{grasp\_force} > 0.5) \cdot \left( 1 - \tanh\left( \frac{\sum_{i=1}^{2} \text{finger\_pos}[i]}{0.1} \right) \right)$$

where grasp_force represents the force exerted by the fingers on the object. finger_pos represents the joint positions of the robot's fingers.

- **Lift End Effector (EE)**

$$R_{\text{lift\_ee}} = \frac{\text{clamp}(\mathbf{p}_{\text{ee},z} - 0.6, 0.0, 0.3)}{0.3} \cdot \mathbb{1}(\text{grasp\_force} > 0.5)$$

where $\mathbf{p}_{\text{ee},z}$ represents the height of the end-effector above the ground.

- **Lift Object**

$$R_{\text{lift\_object}} = \mathbb{1}(\mathbf{p}_{\text{object},z} > z_{\text{init}} + h_{\min}) \cdot \frac{\text{clamp}(\mathbf{p}_{\text{object},z} - z_{\text{init}} - h_{\min}, 0.0, 0.2)}{0.2}$$

where $\mathbf{p}_{\text{object},z}$ represents the height of the object above the ground. $z_{\text{init}}$ is the initial height of the object. $h_{\min} = 0.02$ is the minimal height above the initial height required to achieve the goal.

Details of our weight design for the reward function can be found in Table 7

Table 7: Reward Function Design

| Reward Term | Weight |
|---|---|
| Fingers Open | 5.0 |
| Reaching Object | 15.0 |
| Object Goal Tracking | 80.0 |
| Object Goal Tracking (Fine-Grained) | 70.0 |
| Action Rate Penalty | -1e-4 |
| Joint Velocity Penalty | -1e-4 |
| Contact Forces | 10 |
| Close Fingers | 100 |
| Lift End Effector (EE) | 20 |
| Lift Object | 100 |

## E.2 DATA GENERATION DETAILS

In data generation, as shown in Fig. 2, we feed the images from the $n_{cam}$ first-person cameras of the robot to YOLO-World at each timestep to detect the $n_{obj}$ target objects to be manipulated, thus obtaining $n_{cam} * n_{obj}$ bounding boxes. Then we normalize the bounding box, concatenate them, and obtain a vector $o$ of dimension $4 * n_{cam} * n_{obj}$ as the visual feature input for the control policy. We then store the visual features $o_t^{vis}$, all joint positions of the robot $o_t^{rob}$, and actions $a_t$ for each step in the form of trajectories.

To reduce visual interference between parallel environments and minimize the sim-to-real gap, we incorporate a white room during data collection. The room is designed to be spacious enough to accommodate the maximum working space, including robot, table, and apple. Three RGB-D cameras[3], calibrated with intrinsic and extrinsic parameters matching those of the real robot, capture visual information. During inference, the images from these cameras are processed by YOLO-World to generate object bounding boxes.

Table 8: Camera Parameters

| Camera | Link | Image Shape and Type |
|---|---|---|
| **cam_high** | fr_link6 | Shape: $[480, 640, 4]$
RGB, Range: 0-255 (torch.uint8) |
| **cam_left_wrist** | fl_link6 | Shape: $[480, 640, 4]$
RGB, Range: 0-255 (torch.uint8) |
| **cam_right_wrist** | fr_link6 | Shape: $[480, 640, 4]$
RGB, Range: 0-255 (torch.uint8) |

Given the high memory demands of image data, we run 16 parallel environments on a single GPU [4] for data generation, achieving up to 36k successful trajectories per day. For ACT training, these trajectories are subsequently stored in an HDF5 data structure[5] (see Table 9). In our approach, we convert images to bounding boxes using YOLO-World and store the results in a dictionary format. Each trajectory consists of multiple steps, with each step containing bbox, qpos, and action data.

## E.3 STUDENT POLICY DETAILS

We use RNN and ACT to train our student policy. The details of training parameters can be found in Table 10 and Table 11.

---

[3]But we only use RGB information.

[4]For some 24 GB GPU.

[5]Video-based dataset

Table 9: HDF5 Data Structure

| Group/Attribute | Dataset/Attribute Name | Description |
|---|---|---|
| **Root Attributes** | sim | Boolean indicating simulation (False) |
| | compress | Compression flag from 'CollectEpsBuf' |
| **Datasets** | action | Array of actions taken |
| | reward | Array of rewards received |
| | base_action | Array of base actions |
| | base_action_t265 | (Optional) Base actions from T265 sensor |
| **Observations: states** | qpos | Array of joint positions |
| | qvel | Array of joint velocities |
| | effort | Array of joint efforts |
| **Observations: images** | cam_high | Byte-encoded images from 'cam_high' |
| | cam_left_wrist | Byte-encoded images from 'cam_left_wrist' |
| | cam_right_wrist | Byte-encoded images from 'cam_right_wrist' |

Table 10: RNN Training Parameters

| Parameter | Value |
|---|---|
| lr | 0.002 |
| lr_backbone | 7e-05 |
| epochs | 50 |
| warmup_ratio | 0.1 |
| use_scheduler | cos |
| weight_decay | 0.0001 |
| loss_function | l1 |
| hidden_dim | 512 |
| rnn_layers | 3 |
| rnn_hidden_dim | 512 |
| actor_hidden_dim | 512 |
| policy_class | RNN |
| gradient_accumulation_steps | 1 |
| random_mask_ratio | 0.3 |

### E.4 SIM2REAL TECHNIQUE

We employ the student policy for sim-to-real transfer, where the inputs to the policy included proprioception $o_t^{rob}$ which consists of joint positions, and the visual input $o_t^{vis}$, represented by bounding boxes. The output action is the desired joint position for the next step. Thus, the primary sim-to-real gaps exist in the joint position and bounding box dimensions.

For joint positions, we develop a real-to-sim code that maps the joint positions obtained from teleoperation on a real-world robot to a simulated robot. We evaluate whether the actions performed at specific target positions were consistent between the real-world and simulated environments. For example, when teleoperating to grasp an apple at a precisely measured location, both the real-world and simulated apples are set at the same coordinates. We used the main arm of the real-world robot to observe whether the simulated robot could successfully grasp the apple.

Special handling is required for the gripper, as there are discrepancies between the URDF model of the gripper and its real-world counterpart. In the simulation, the gripper has two joints, while in reality, it functions as a single joint, leading to differences in value interpretation. To address this, we collect 10 data sets for fitting and utilize function interpolation to map the simulated gripper joint positions to those of the real-world gripper. This approach ensures that the state space and action space are aligned between the simulation and the real environment.

The sim-to-real gap for the bounding boxes corresponds to the image sim-to-real gap. We perform hand-eye calibration on the camera to ensure that the cameras in both the simulation and the real-

Table 11: ACT Training Parameters

| Parameter | Value |
|---|---|
| num_epochs | 1000 |
| lr | 4e-05 |
| lr_backbone | 7e-05 |
| weight_decay | 0.0001 |
| kl_weight | 10 |
| backbone | resnet18 |
| position_embedding | sine |
| loss_function | l1 |
| chunk_size | 32 |
| hidden_dim | 512 |
| dim_feedforward | 3200 |
| enc_layers | 4 |
| dec_layers | 7 |
| nheads | 8 |
| dropout | 0.2 |
| train_loader_len | 23 |
| warmup_ratio | 0.1 |
| scheduler | cosine |

world robot are approximately aligned. For finer adjustments, we select a specific point in front of the robot and a specific-sized object, observing the bounding boxes obtained from both the simulator and the real-world camera. Since we use bounding boxes, other factors such as lighting and background variations in the sim-to-real gap are not a concern. Ultimately, this method of adjustment ensures that the intrinsic and extrinsic parameters of the camera are consistent between the simulation and the real-world environment.

Other methods to ensure successful Sim2Real, such as domain randomization, are described in the previous section (Sec. 3.2).

## F  HARDWARE DETAILS

The robot is equipped with a dual-arm design, providing a total of 14 degrees of freedom (DoF), with 7 DoF per arm for versatile movement. Each arm has a maximum valid payload of 1500g, and the gripper has a range of up to 80mm, allowing the robot to perform a variety of grasping and manipulation tasks efficiently. The detailed hardware information and an image of the robot can be found in Figure 22 and Table 12.

| Parameter | Value |
|---|---|
| Size | $1080 \times 700 \times 1140$ |
| Arm weight | 4.2 kg |
| Arm Payload | 3000 g (peak) |
| | 1500 g (valid) |
| Arm reach | 600 mm |
| Arm repeatability | 1 mm |
| Arm working radius | 653 mm |
| Joint motion range | J1: ±154°, J2: 0°~165° |
| | J3: -175°~0°, J4: ±106° |
| | J5: ±75° , J6: ±100° |
| Gripper range | 0-80 mm |
| Gripper max force | 10 NM |
| DoF | $7 \times 2 = 14$ |

Table 12: Technical specifications.

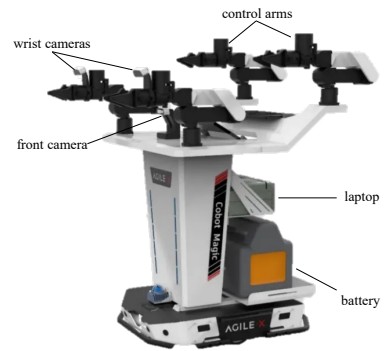

Figure 22: Hardware features.

# G    LIMITATIONS AND DISCUSSIONS

One of the limitations of ManiBox is that the accuracy of the obtained bounding box depends on the capability of the preceding visual model. As the YOLO-World sometimes may not be able to accurately partition the bounding box, ManiBox may grasp the wrong positions due to the wrong input bounding box. Also, the bounding box is mainly for objects with convex shapes and may be insufficient for handling flexible objects and fluids. In the future, a promising direction is to apply more powerful visual models like Grounding DINO and capture more semantic visual signals like segmentation via SAM in ManiBox to improve its ability. Also, for some complicated manipulation tasks like folding up clothes, directly training an RL teacher agent in the simulator may be difficult. Thus how to utilize human demonstration or state machine to get the expert teacher agent is also worth researching.

