# OpenReview forum: "ManiBox: Enhancing Spatial Grasping Generalization via Scalable Simulation Data Generation"
_ICLR.cc/2025/Conference — Submitted to ICLR 2025_

### Official Review · Reviewer_rbGJ · 2024-10-15

**Soundness:** 1
**Presentation:** 2
**Contribution:** 1
**Rating:** 3
**Confidence:** 3

**Summary:**

This paper introduces a manipulation method called ManiBox, which operates using bounding-box inputs. The policy is trained using a student-teacher framework. Specifically, a teacher policy is trained with PPO using privileged information, such as the object's 3D position. A dataset of expert trajectories is then generated from this teacher policy, which is subsequently used to train a student policy through imitation learning. Isaac Lab is employed as the simulator to facilitate scalable data collection. The authors demonstrate that as the workspace size increases, a larger amount of expert data is required to train effective student policies. Finally, the trained student policies can be deployed on a real robot by using bounding boxes generated from an open-vocabulary object detection model.

**Strengths:**

**General Idea**
-  Leveraging vision-foundation models to train policies that can be deployed on real robots is an interesting and relevant approach for addressing many real-world manipulation tasks.

**Real-robot Deployment**
- The proposed method is tested on a real robot, verifying that the inputs from the open-vocabulary detection model can be used to transfer the learned behaviors.

**Visual Presentation**
- The overview figures, particularly Figure 2, effectively clarify the proposed model.

**Weaknesses:**

**Low Novelty**
- The paper does not introduce a novel method, but rather employs a specific student-teacher learning formulation, using bounding boxes in the student's input space. Additionally, the finding that the amount of expert data required to train a proficient student policy scales with the robot's workspace size is unsurprising.
- The method trains on a single object (with randomization in its scale) using a parallel gripper in an uncluttered tabletop scenario, which is a very simple task configuration that has been addressed without the need for interactive RL policies (see e.g. https://inria.hal.science/inria-00325794/document)

**Method Limitations**
- Representing objects through 2D bounding boxes provides only a rough approximation of an object’s convex hull, which is likely insufficient for generalizing to objects with diverse shapes.
- The method currently relies on multiple camera observations. Inferring object states from a single RGB-D camera or using the history of wrist-camera observations from different poses would make the approach easier to deploy.

**Results and Research Claims**
- The Abstract claims that the paper will demonstrate that most models’ spatial generalization challenges stem from high data requirements for spatial understanding. However, this is not substantiated, as dataset size is the only variable that is investigated in the experiments. Moreover, reformulating the problem from joint-space actions and observations to end-effector control, with object position relative to the robot's gripper, could reduce variability that is unnecessary for task completion.
- Object generalization is evaluated by testing on items different from those encountered by the teacher policy. It is mentioned that two objects are out of distribution, yet since the teacher policy training does not cover these objects, there is no reason to expect it to generalize to this configuration. Furthermore, details on how this experiment was conducted—such as the number of trials and object positions—are missing. While it is possible to learn a grasping strategy that is sufficiently general to cover the tested items, this is not a feature of the method since the teacher policy is never incentivized to do so.
- In the Background Generalization results section, the model's generalization ability is attributed to its integration of historical information, multi-camera data, and random masking, but none of these claims are verified.

**Questions:**

- Why is the teacher policy used to generate a dataset of expert trajectories from which the student policy is learned, instead of using DAgger, given that the teacher policy could be queried for expert actions at each state?
- Why does the teacher policy not have access to the ground-truth object size? If the student policy can infer the object size from detected bounding boxes, the teacher policy could also utilize this information to adapt its grasp for different object sizes.
- Following line 255, it is mentioned that the state space dimensionality is high, while one of the motivations for using student-teacher learning was to formulate a compact input space for the teacher policy to facilitate efficient learning. Furthermore, it is claimed that this reduces the exploration space of the RL environment, but the details are missing in Appendix B.1. Could you explain how the exploration space was reduced and why?
- In Figure 2, you state there is a bijective mapping between the privileged object representation and its bounding-box detections, implying that the inputs required to deploy the teacher policy could be computed directly from visual detections. If that is the case, why is the student-teacher framework necessary?
- If the model controls the robot via joint commands, how are self-collisions and collisions with the tabletop avoided?
- The data in Figure 3, showing the scaling relationship between spatial generalization and the amount of data, appears to deviate significantly from the expected trend, particularly in the 10cm x 10cm x 10cm workspace. Could you explain why this occurs, and whether the data points were reported for a single seed?

---

> ### Author Response · Authors · 2024-11-22
>
> Thank you for your insightful and constructive comment. We greatly appreciate your acknowledgment of the general idea, real-robot deployment, and visual presentation of our work. Below, we address each of your concerns in detail.
>
> **W1.1**: Novelty of our approach
>
> **A**: While building on the student-teacher framework, our work introduces the following contributions that distinguish it from prior methods:
> - Bounding-box-guided spatial generalization: By incorporating bounding boxes into the student's input space, we address the high-dimensional limitations of visual generalization. This improvement enables adaptability across diverse spatial positions, objects, and backgrounds.
> - Discovery of spatial generalization laws: We are the first to identify **two spatial generalization laws** relating workspace size, data volume, and policy performance. These laws provide actionable insights for estimating data requirements and have been referenced by peers in embodied AI.
> - Efficiency and practicality: Our method, ManiBox, achieves strong spatial and visual generalization with minimal computational resources. For example, training requires only 2 hours for the teacher policy, 1 day for data generation, and 2 minutes for the student policy (single GPU, RTX 4090).
>
> **W1.2**: Relationship between workspace size and data requirements
>
> **A**: While the overall trend may appear intuitive, our work provides the **first empirical quantification** and reveals unexpected insights:
> - Non-linear scaling: Data requirements grow sublinearly with workspace size, following a power law (Figures 4, 15-16). This efficiency offers practical guidance for scaling.
> - Saturation effect: Success rates exhibit Michaelis-Menten kinetics relative to data volume, showing diminishing returns as data increases (Figure 3). These findings deepen the understanding of spatial generalization.
>
> **W1.3**: Experimental setup and scalability
>
> **A**: We agree that more challenging scenarios are essential to validate robustness. The "Complex Generalization" experiments in Figure 5 already feature multi-object and cluttered environments, such as non-tidy desktops with computers and other objects, highlighting ManiBox's strong generalization capabilities.
>
> To further support this, we conducted additional experiments on manipulation tasks like **"pouring water"**, "grabbing the handle of a cup", "cluttered table", "messy table" (see Appendix C and Figure 13-14), demonstrating ManiBox's versatility beyond standard grasping tasks.
>
> Compared to prior works like AnyGrasp or the Inria paper, which focus solely on grasping, ManiBox generalizes across diverse task types and settings using its bounding-box-guided policy learning approach. Moreover, ManiBox achieves strong results efficiently, requiring minimal computational resources, making it highly practical for real-world deployment.
> We have updated the manuscript to include these additional experiments and discussions.
>
> **W2.1**: Representation with 2D bounding boxes
>
> **A**: We acknowledge that representing objects with 2D bounding boxes offers only a rough approximation of an object's convex hull, which may present challenges for highly irregular or diverse shapes. However, our method prioritizes simplicity and computational efficiency, making bounding boxes a practical choice for grasping and simple manipulation tasks while enabling exploration of spatial generalization laws.
>
> To address this limitation, ManiBox can flexibly integrate other object detection models. For example, in newly added tasks like **"grabbing the handle of a cup"** (see Appendix C), Grounding DINO enables detailed detection of specific parts. Similarly, SAM can provide finer segmentation, which we plan to explore in future work to improve generalization to complex object geometries.
>
>
> **W2.2**: Reliance on multiple camera observations
>
> **A**: As shown in Figure 2, the number of cameras does not increase the deployment complexity, as it only impacts the dimensionality of the bounding box representation. Using multiple camera observations enhances policy robustness compared to a single-camera setup. Interestingly, biological intelligence, such as human vision, also benefits from binocular perception, which inspires our approach.

---

> > ### Author Response · Authors · 2024-11-22
> >
> > **W3.1**: Clarify the claim about spatial generalization challenges
> >
> > **A**: Our choice to focus experiments on dataset size was intentional, aiming to isolate the impact of data requirements on spatial understanding, a critical factor for real-world applications. This approach draws inspiration from the formula (4.1) in OpenAI's Scaling Laws paper (https://arxiv.org/pdf/2001.08361), which shows that a model's performance $L(N,D)$ depends independently on both the number of parameters $N$ and the amount of data $D$. This independence highlights the importance of systematically exploring spatial generalization as a function of dataset size, which our work addresses.
> >
> > While the problem was not explicitly reformulated to end-effector control with relative object positioning, as suggested by the reviewer, this approach could reduce unnecessary variability. Future work will explore such strategies to **enhance spatial generalization while reducing data complexity**.
> >
> > **W3.2**: Object generalization
> >
> > A: We would like to clarify that the teacher policy is trained exclusively **on Apples of varying sizes** to facilitate bounding-box-guided data generation and student policy learning. Notably, the student policy successfully **adapts to several bounding box sizes** (rather than two distinct objects) that are out-of-distribution from its generated training set, demonstrating the spatial generalization capability of our ManiBox.
> >
> > The ability of the student policy to generalize to various objects stems from the bounding-box-guided teacher-student policy distillation process. This process enables the student to learn to **grasp the center of an object's bounding box** after the targeted object and its box are identified by advanced open-vocabulary detection models.
> > Details on our experimental setup, including the number of trials and object positions, are provided in Appendix D.1 and D.3.2 for reference.
> >
> > **W3.3**: Background generalization
> >
> > A: To ensure rigor, we have clarified in the manuscript that the model's background generalization ability is attributed to the integration of historical information, random masking, and the bounding-box-guided policy. Detailed explanations and supporting arguments, including an ablation study comparing models trained on selected timesteps versus full trajectories, are provided in **Appendix B**. Additionally, to further illustrate the policy's robust inference capabilities, we tested its performance under significant visual uncertainty. During inference, we introduced random masks and noise to the visual model’s bounding box outputs, demonstrating the policy’s ability to generalize effectively in such scenarios.
> >
> > **Q1**: Why is the teacher policy used to generate a dataset of expert trajectories instead of using DAgger?
> >
> > A: While DAgger allows querying the teacher policy for expert actions at every state during training, our decision to use pre-generated expert trajectories was motivated by the following considerations:
> >
> > - **Computational efficiency**: Repeated querying of the teacher policy in DAgger can be computationally expensive. In real-world scenarios, the teacher policy can't work without privileged information. In simulator, DAgger requires simulator rendering and detection of model inference during data generation, which can severely slow down student policy training. Pre-generating trajectories minimizes this overhead, enabling efficient training. For example, our framework requires only 2 hours for teacher policy training, 1 day for data generation, and 2 minutes for student policy training on a single GPU (RTX 4090).
> >
> > - **Offline training feasibility**: Pre-generated trajectories allow offline training of the student policy, which is advantageous when deployment environments or hardware constraints limit online querying during training.
> >
> > - **Consistency for generalization**: Our approach emphasizes bounding-box-guided policy distillation. Pre-generated trajectories ensure consistent data quality, enabling the student policy to focus on **learning robust spatial generalization** without the additional complexity of dynamic querying.
> >
> > While DAgger has theoretical advantages in reducing compounding errors through interleaved expert guidance, our experiments demonstrate that pre-generated datasets are sufficient to achieve strong generalization in ManiBox (see Section 4). In future work, we plan to explore combining the efficiency of pre-generated data with on-the-fly querying to further enhance performance.

---

> > > ### Author Response · Authors · 2024-11-22
> > >
> > > **Q2**: Why does the teacher policy not have access to the ground-truth object size?
> > >
> > > A: Our teacher policy already **utilizes the ground-truth object center** (x,y,z) provided by the simulator and learns to grasp the center of an object through reinforcement learning. This design ensures that the teacher policy generates highly optimal expert actions, as demonstrated by a 100% success rate in our simulator.
> > >
> > > In contrast, the student policy is designed to learn to grasp the center of an object's bounding box instead of the precise object center. This relaxation from a point (object center) to a bounding box is intentional and aims at **facilitating sim-to-real transfer and real-world deployment**, where precise ground-truth object centers are often unavailable.
> > >
> > > We believe this distinction between the teacher and student policy enables the robustness and practicality required for real-world applications, while maintaining high performance during the distillation process.
> > >
> > > **Q3**: How is the exploration space reduced and why?
> > >
> > > A: The exploration space is reduced by leveraging the teacher-student framework, where the teacher policy uses a simplified state representation instead of high-dimensional image inputs. We have clarified this in Section 3.2, "Teacher Policy Training."
> > >
> > > The reduction in exploration space is achieved through three key mechanisms:
> > > - The teacher policy **relies on robot and object-specific information rather than raw visual inputs**, significantly reducing the dimensionality of the state space.
> > >
> > > - **Privileged information and tailored reward functions** focus the policy on critical steps of the task, narrowing the exploration space to regions most relevant for successful task completion.
> > >
> > > - The simplified state representation allows for **efficient parallelization**, enabling **8092** parallel environments during training compared to **24** in visual RL, further enhancing the policy's ability to explore effectively.
> > >
> > > These mechanisms ensure that exploration is more targeted and computationally efficient, enabling faster convergence and improved performance.
> > >
> > > **Q4**: Why is the student-teacher framework necessary if the teacher policy inputs can be computed directly from visual detections?
> > >
> > > A: The student-teacher framework is essential for three key reasons:
> > >
> > > - **Efficient training**: Training the teacher policy with privileged object information avoids the need for visual inputs, enabling large-scale RL training without the computational overhead of visual modules.
> > >
> > > - **Simplified deployment**: The student policy adapts the teacher's knowledge for real-world scenarios (sim-to-real), using bounding boxes as inputs instead of privileged information, which is unavailable during deployment.
> > >
> > > - **Robustness**: Training the teacher policy directly with bounding boxes would result in degraded performance under real-world visual inaccuracies. The student-teacher framework ensures a robust transfer of the student policy from simulation to reality.
> > >
> > > This approach balances efficient training with practical, robust deployment.
> > >
> > > **Q5**: How are self-collisions and collisions with the tabletop avoided when using joint commands?
> > >
> > > A: Collisions are avoided through the following mechanisms:
> > >
> > > - **Tabletop collisions**: Since objects are always on the table, their bounding boxes implicitly encode the contact position with the tabletop. This ensures that the robot avoids collisions with the table when interacting with objects.
> > >
> > > - **Self-collisions**: Using joint position control (instead of end-effector position control) eliminates the risk of singularities and self-collisions.
> > >
> > > - **Reward design**: In teacher policy learning, our reward function incorporates **action rate penalty** and **joint velocity penalty**, which strongly penalize trajectories with high joint velocities typically associated with collisions. This discourages collision-prone behaviors during training.
> > >
> > > - **Data filtering**: During data generation, any trajectory involving a collision is marked as unsuccessful and filtered out, ensuring only collision-free data is used for student training.
> > >
> > > **Q6**: Explanation for the deviation in the 10cm x 10cm x 10cm workspace in Figure 3
> > >
> > > A: Thank you for pointing this out. In the original Figure 3, the slight deviation of the two blue points (at 1000 and 4000 trajectories) in the 10cm workspace was due to testing variance. To address this, we **updated Figure 3 by testing with three random seeds**, which has significantly stabilized the results for the 10cm workspace. The updated figure now better reflects the scaling relationship with reduced variability.

---

> > > > ### Comment · Reviewer_rbGJ · 2024-11-25
> > > > **Thank you for your reply**
> > > >
> > > > I would like to thank the authors for their detailed response to my questions and concerns. While most of my questions were about the motivations for certain design choices, and these have been addressed in the response, my main concern remains about the novelty of the work and the validation of the research claims.
> > > >
> > > > Regarding object generalization, you mention that the policy was trained on apples of different sizes. What is the motivation for this, and why would this lead to good generalization? A policy trained on a set of diverse objects would be encouraged to learn more generalizable strategies than training on scaled versions of a single object. While always grasping at the center of an object's bounding box seems to work for the objects studied, this behavior is unlikely to transfer to objects of very different shapes and sizes.
> > > >
> > > > Regarding the argument for behavior cloning over DAgger, I still believe that DAgger is the more natural formulation for this student-teacher learning problem. The computational overhead of querying a teacher policy on a low-dimensional observation is very small.
> > > >
> > > > Regarding the observation space of the teacher policy, which uses only object positions and not their size, you say "This design ensures that the teacher policy generates highly optimal expert actions". How does this design ensure this? Furthermore, the performance of the student policy that you eventually use is upper bounded by the performance of the teacher policy that you train in the first stage. Because the teacher policy does not receive information about object shape or size, it cannot adapt its behavior to generalize to different objects based on that information. Therefore, the information about object shape and size provided to the student policy is irrelevant, since it is trying to imitate the teacher policy and no generalization based on this information can occur.
> > > >
> > > > Regarding the necessity of the student-teacher learning framework. First, using the detected bounding boxes to estimate the center of the object would make the teacher policy usable in the real world. Second, one could also directly train the policy in simulation on approximate bounding box observations, for example by sampling points on the object mesh and projecting them into the image space of the desired camera and then computing the bounding box around them.
> > > >
> > > > Regarding collisions of the robot with itself and the table, you write: "Since objects are always on the table, their bounding boxes implicitly encode the contact position with the tabletop. This ensures that the robot avoids collisions with the table when interacting with objects." How does the implicit encoding of the contact position with the tabletop **ensure** this?
> > > >
> > > > I want to thank the authors again for their contribution and the detailed answer, but I will keep my score.

---

> > > > > ### Author Response · Authors · 2024-11-26
> > > > > **We hope this provides clarity and clears up any misunderstandings between us, and welcome any further questions.**
> > > > >
> > > > > Thank you for raising these thoughtful concerns. This work primarily focuses on **exploring spatial generalization laws** through **efficient offline data generation**. Our framework reflects deliberate engineering to achieve these goals, and we have also tried our best to validate policy generalization across objects and backgrounds to demonstrate practical applicability. We hope this provides clarity and clears up any misunderstandings between us, and welcome any further questions.
> > > > >
> > > > > ## Regarding Object Generalization ##
> > > > > A: **Why we chose apples:** We choose apples for their flexibility in being **scaled into a variety of shapes (e.g., elongated, flattened, or cylindrical forms)**. This design allows the policy to efficiently learn adaptive grasp behaviors through RL, without repeatedly designing or modeling various objects. Crucially, our approach aims to **learn a grasping policy with enhanced spatial generalization using low-dimensional bounding-box-guided data generalization and policy distillation**, where the specific object type used in the teacher policy is less important. In fact, "apples" could easily be replaced with other regular objects to achieve the same results.
> > > > >
> > > > > **Generalization principle:** Our reasoning aligns with a key principle in robotic manipulation: **many complex rigid bodies can be decomposed into combinations of smaller links and joints**. Similar to MuJoCo or URDF modeling, where rigid objects are represented as assemblies of simpler components (e.g., links composed of cylinders and half-spheres), our method leverages these fundamental shapes. By training the policy to grasp such basic forms of various sizes, our approach can generalize effectively to irregular objects by detecting and targeting specific parts (e.g., handles or links) for precise manipulation.
> > > > > We conducted additional experiments on complex tasks such as "pouring water" (applying our approach to multi-object manipulation tasks), "grabbing the handle of a cup"(applying our method to the grasping of detailed parts of irregular objects), and "cluttered table" to validate ManiBox's robustness and scalability (Appendix C, Figures 10-14, and **Supplementary Material**).
> > > > >
> > > > > **Addressing visual discrepancies:** While training on a diverse set of **real-world objects may lead to broader generalization** (e.g., RT-1), our simulation-based approach focuses on overcoming the gap between simulation and reality due to **high-dimensional visual discrepancies**. Methods like ACT, which train on diverse simulated objects, often struggle with sim-to-real transfer, as demonstrated by their low success rates in Table 1 of the paper.
> > > > >
> > > > > ## Regarding Policy Generalization Capabilities ##
> > > > > A: We validated this claim with two additional experiments, **"Grabbing the Handle of a Cup"** and **"Pouring Water"**, detailed in Appendix C and supplementary material:
> > > > > - In "**Grabbing the Handle of a Cup**", the policy generalized to grasp fine-grained parts, such as the handle of a cup, after training with diverse bounding boxes.
> > > > > - In "**Pouring Water**", we demonstrated that adjusting the teacher policy enables the execution of more complex manipulation tasks, showcasing the versatility of our approach.
> > > > >
> > > > > These results demonstrate that training with a diverse set of bounding boxes rather than diverse objects enables the policy to generalize effectively to specific parts of irregular objects and handle complex tasks.

---

> > > > > > ### Author Response · Authors · 2024-11-26
> > > > > > **We hope this provides clarity and clears up any misunderstandings between us, and welcome any further questions.**
> > > > > >
> > > > > > ## Behavior Cloning over DAgger ##
> > > > > > A: While DAgger mitigates compounding errors, our results show no significant limitations with pre-generated data in our framework. Instead, our approach emphasizes **efficiency and scalability**, which are essential for large-scale experiments:
> > > > > >
> > > > > > - **Flexibility for experimentation:**
> > > > > >   - Pre-generated data enables **2-minute student policy training**, allowing us to **explore dozens of training methods (e.g., hyperparameter tuning, architecture adjustments, random mask) within an hour**. We tried dozens of different training methods, and each of these student policies training methods can be trained and tested quickly.
> > > > > >   - DAgger would require over a day for each training run, making iterative experimentation infeasible. For example, if we use DAgger, dozens of training methods take more than a hundred days to iterate.
> > > > > >
> > > > > > - **Efficiency in parallelization:**
> > > > > >    - DAgger requires the simulator, detection module, and student training to run concurrently, increasing GPU memory usage and reducing the number of parallel environments, thus slowing down data generation.
> > > > > >    - Decoupling these stages allows our pipeline to fully utilize GPU resources, pairing simulator rendering with detection inference for maximum speed. This optimization reduces the primary bottleneck: **data generation (1 day)**.
> > > > > >
> > > > > > This decoupled design ensures **optimal resource utilization and rapid iteration**, making it more practical for our large-scale spatial generalization study.
> > > > > >
> > > > > > Moreover, the performance of BC in our approach is sufficient and does not raise significant distribution bias problems, e.g., the simulation experiment reaches a 90% success rate when the data volume is sufficient (in Figure 3), and the success rate of the real robot in Table 1 is also around 100%. In future work, combining DAgger and BC can be explored to further improve the performance.
> > > > > >
> > > > > > ## Regarding the Observation Space of the Teacher Policy ##
> > > > > > A:
> > > > > > - Please refer to the fact mentioned earlier that **"complex rigid bodies can be decomposed into combinations of smaller links and joints."**
> > > > > > - The teacher policy is trained with objects of varying sizes and positions. During training, the teacher policy receives inputs related to the objects, such as contact forces and object positions. These inputs guide the teacher policy to successfully grasp objects of different sizes and positions through RL.
> > > > > > - The teacher policy learns to handle regular objects of various sizes effectively. Subsequently, the student policy, combined with a detection model, identifies detailed features of irregular objects. This allows the student policy to grasp the intricate parts of irregular objects.
> > > > > > - As for determining which specific detailed part of an irregular object should be grasped, this prompt can be generated using a large language model with common sense.
> > > > > >
> > > > > >
> > > > > > ## Regarding the Necessity of the Student-Teacher Learning Framework ##
> > > > > > A: Our teacher-student framework is specifically designed to address key challenges in both simulation and real-world deployment:
> > > > > >
> > > > > > - **Robustness to detection errors:** Using bounding boxes directly to estimate the object center can introduce **estimation errors**, particularly when the detection model fails to identify objects in certain timesteps (a common issue in real-world scenarios). **The teacher policy lacks historical context and cannot recover from such failures**, leading to cascading errors in subsequent timesteps. By contrast, our student policy is designed to handle raw bounding box inputs, **leveraging temporal information through its recurrent architecture**, ensuring robust performance even with incomplete or noisy detections.
> > > > > >
> > > > > > - **Efficiency in simulation:**
> > > > > >     - Training a policy directly on bounding box observations in the simulator **requires rendering camera views and running detection models**, which drastically reduces the simulation step speed. This inefficiency makes it impractical to generate large-scale datasets. Our decoupled architecture allows the teacher policy to leverage privileged information (e.g., object position and size) to generate trajectories efficiently, achieving a workflow of **1 day for data generation and 2 minutes for student training**.
> > > > > >     - As mentioned earlier, since the student policy training time is only around two minutes, the decoupled framework allows us to quickly try a large number of student policy training techniques.
> > > > > >
> > > > > >
> > > > > > ## Regarding Collisions with the Table ##
> > > > > > A: The bounding box includes the z-axis information of the object's base. When the object is on the table, this z-value effectively represents the tabletop's position. Since our policy is trained to grasp the object’s center, it naturally avoids moving below the object’s base, preventing collisions with the table.

---

> > > > > > > ### Author Response · Authors · 2024-11-29
> > > > > > > **Follow-Up on Our Latest Response**
> > > > > > >
> > > > > > > Dear Reviewer rbGJ,
> > > > > > > We would like to kindly remind you that we have submitted our latest response addressing your concerns. Additionally, we have included new comparison experiments with heuristic approaches, such as the Inverse Kinematics algorithm from Isaac Lab. Our experiments show that while these methods are prone to failure—especially at the workspace edges—our RL-based approach performs robustly across a wider range of scenarios, as demonstrated by the following success rates:
> > > > > > >
> > > > > > > | Method         | Success Rate of Full Space |
> > > > > > > |----------------|----------------------------|
> > > > > > > | Inverse Kinematics (IK)  | 0.6875 ± 0.0510       |
> > > > > > > | Key Steps      | 0.0625 ± 0.0884       |
> > > > > > > | Without Random Mask   | 0.5833 ± 0.0780       |
> > > > > > > | **Student Policy of ManiBox (Ours)**           | **0.9167 ± 0.0295**       |
> > > > > > >
> > > > > > > - **Inverse Kinematics (IK)**: This approach directly uses the ground truth object pose as the target end-effector (eef) pose, then computes the joint positions via IK to move the robot arm to the target. However, we observe that IK performs poorly at the edges of the workspace, especially when objects are near the table surface or at low positions. Examples of IK failures when the table is very low can be seen on this **[anonymous link](https://anonymous.4open.science/r/iclr2025-3C2C/README.md)**. The reason for IK failures is possible:
> > > > > > >     - The robotic arm may approach singular configurations where the Jacobian matrix becomes singular, causing unstable or undefined IK solutions.
> > > > > > >     - Near the workspace edge, numerical inaccuracies may result in imprecise IK solutions. Moreover, approximation methods (e.g., pseudoinverse) may struggle to converge to a valid solution in complex boundary cases.
> > > > > > > - **Key Steps**: In this approach, only key steps of the trajectory are used to model the grasping task.
> > > > > > > - **Without Random Mask**: This variant trains the student policy without random masking for object positioning.
> > > > > > >
> > > > > > > Key Insights:
> > > > > > > - **IK Limitations:** As expected, IK struggles in the workspace boundaries due to singularities, which cause failures when the object is positioned at the edges of the table or at low heights. This limitation is inherent to traditional geometric methods like IK, which rely on precise pose knowledge and are prone to errors in challenging environments.
> > > > > > > - **Our RL-Based Approach**: In contrast, **ManiBox (Ours)** utilizes RL to learn robust grasping teacher policies by directly planning the joint positions, enabling successful manipulation across a wide range of object positions. Our method does not rely on explicit pose knowledge and is highly effective even at challenging object locations that cause IK to fail.
> > > > > > >
> > > > > > > We hope this extended comparison clarifies the advantages of our approach, particularly in terms of robustness and generalization. Moreover, as previously mentioned, **We conducted some complex tasks such as "pouring water" (applying our approach to multi-object manipulation tasks) and "grabbing the handle of a cup"(applying our method to the grasping of detailed parts of irregular objects(Appendix C, Figures 10-14, and Supplementary Material)** to demonstrate the scalability of our approach. If there are any additional questions or points needing clarification, please do not hesitate to reach out. Thank you again for your time and feedback. We look forward to your insights.
> > > > > > >
> > > > > > > Best,
> > > > > > > Authors

---

### Official Review · Reviewer_4x8i · 2024-10-30

**Soundness:** 2
**Presentation:** 2
**Contribution:** 2
**Rating:** 6
**Confidence:** 3

**Summary:**

The paper introduces ManiBox, a framework for improving the generalization of robotic grasping through bounding-box-guided manipulation. Using a teacher-student policy setup, the teacher generates data in simulation, and the student learns to transfer this knowledge to real-world scenarios. Objects are represented with bounding boxes, aiming to reduce complexity and enhance generalization. A key finding is the scaling law that shows data requirements grow non-linearly with spatial volume. While the system demonstrates strong Sim2Real performance, some design choices and experimental setups leave room for clarification.

**Strengths:**

**Effective Sim2Real Transfer:**

The random masking strategy helps the system transfer successfully from simulation to real-world environments.

**Background Generalization:**

The system maintains good performance across diverse backgrounds, which adds practical robustness.

**Scaling Law Insight:**

The identified relationship between data volume and spatial generalization offers valuable guidance for data-driven models.

**Weaknesses:**

**Limited Object Diversity:**

The experiments use similar and simple objects, which limits the demonstration of the full generalization potential. A wider variety of objects might better highlight the benefits of the approach.

**Unclear Bounding Box Usage:**

The relationship between bounding boxes for object detection and manipulation is not clearly explained. This makes it hard to understand how they work together effectively.


**Fixed Cubic Space Constraint:**

The use of fixed cubic spaces (b x b x b) seems arbitrary. It’s unclear if more flexible bounding volumes could improve results.
Complex Camera Setup:

The use of three cameras raises questions, as stereo vision with two cameras is often sufficient for localization. The necessity of the third camera is unclear.

**Fragmented Descriptions of Settings:**

The simulation, real-world, and policy setups are described separately, making it difficult to compare them. A more structured comparison would improve clarity.


**Lack of Vertical Variations:**

The experiments seem limited to flat surfaces. It’s unclear if the system can handle objects with more Z-axis variation or bounding boxes placed in mid-air.

**Formula 4 Confusion:**


Equation 4 lacks clear explanations of some symbols and their meanings, making it harder to follow.

**Questions:**

1. **Can Bounding Boxes Be Scaled or Combined?**
    - Would it be possible to **scale or combine multiple bounding boxes** to handle more complex objects and improve generalization?
2. **Why Use Fixed Cubic Spaces?**
    - Is there a reason for using **b x b x b cubes** as operational spaces, or could other bounding volumes work better?
3. **What Are the Camera Setups in Real-World Experiments?**
    - Are cameras mounted on the **robot arm and the environment**? What role does each camera play?
4. **Is the Bounding Box 2D or 3D?**
    - Are the bounding boxes used **2D projections** or full **3D representations**?
5. **Can Bounding Boxes Exist in Mid-Air?**
    - Could the system define **floating bounding boxes**, or are they constrained to surfaces?
6. **What Background Challenges Were Faced?**
    - Could the paper clarify the **specific challenges** from changing backgrounds and how they were addressed?
7. **What Are the Teacher Policy’s Hardware Requirements?**
    - What specific **hardware resources** are needed for the teacher policy?
8. **Where Are Random Points Placed?**
    - Are the **random points** generated at the **object’s center or across its surface**?
9. **What Are the Simulation Object Types and Task Settings?**
    - A more detailed description of the **simulation tasks and object setups** would help understand the system’s scope.
10. **How Is Trajectory Quality Linked to Rewards?**
    - What criteria are used to **distinguish good trajectories** from bad ones in the reward function?
11. **Need for Clearer Visuals**:
    - Including **bounding box annotations in simulation screenshots** would help readers, especially those without robotics experience, better understand the process.

---

> ### Author Response · Authors · 2024-11-22
>
> Thank you for your thoughtful and positive feedback. We greatly appreciate your recognition of the strengths of our work, including the effectiveness of our sim-to-real transfer strategy, the robustness of background generalization, and the valuable insights provided by the scaling law relationship.
>
> **W1**: Object diversity
>
> A: In our experimental results (Figure 5), we demonstrated that our method performs effectively across a diverse range of objects, including various cups, bottles, and different types of fruits. Additionally, **we have supplemented two new experiments in Appendix C on the "pouring water" and "Grab the handle of the cup" task**, which requires precise manipulation, showcasing the generality of our approach for handling complex objects and tasks.
>
> Furthermore, our method is adaptable to different object detection models. For instance, Grounding DINO can enable detailed detections (e.g., cup handles or door handles), and SAM can provide more refined segmentation. These extensions, planned for future work, will further enhance the scalability of our approach to handle a broader variety of objects and more intricate tasks.
>
> **W2**: Bounding box usage
>
> A: **Bounding boxes are critical for calculating an object's spatial position**. Using object detection models like YOLO-World, we determine the object's center from bounding box data (see Lemma 2) and provide this information to the student policy. The student learns to grasp the center of the bounding box, offering a robust and deployment-friendly solution.
>
> Bounding boxes **provide strong generalization across diverse objects, positions, and backgrounds, while being computationally efficient**. Unlike the ideal privileged object information (e.g., exact object positions) used by the teacher, bounding boxes are practical for real-world deployment and enable rapid student policy training (1 day for data generation, 2 minutes for training on an RTX 4090). As a bridge between ideal object positions and real-world detection, training the student policy with bounding boxes mitigates performance degradation caused by real-world visual inaccuracies.
>
> **W3.1**: Fixed cubic space constraint
>
> A: Our goal is to explore the relationship between spatial generalization and data quantity by examining patterns across different workspace ranges. Based on the robotic arm's estimated maximum working range (41 cm × 30 cm × 28 cm), we systematically expanded the testing spaces from a fixed point to 5 cm, 10 cm, and 20 cm. This progressive expansion allows us to analyze how spatial generalization evolves with increasing workspace size and changing data distribution. Using fixed cubic spaces ensures a controlled and consistent evaluation framework.
>
>
> **W3.2**: Complex camera setup
>
> A: The three-camera setup is based on the default configuration of our robotic system. However, in practice, only two cameras are effectively used for most tasks. For example, the right camera often fails to capture objects during left-arm manipulation. Thus, the third camera primarily acts as a backup and does not introduce additional complexity into the system. The performance and generalization results reported are achieved using two primary cameras for detection and localization.
>
> **W4**: Fragmented descriptions of settings
>
> A: To improve readability, we have included a pseudo-code representation in Algorithm 1, titled "Training and Deployment Algorithm of ManiBox." This outlines the teacher policy training, simulation data generation, student policy training, and real-world deployment. We hope this addition enhances clarity and accessibility for readers.
>
> **W5**: Lack of vertical variations
>
> A: In our experiments (Figure 5), we used a height-adjustable table to create objects with varying Z-axis positions. As the student policy relies on bounding boxes to locate objects, it is capable of handling objects in mid-air. To further validate this, we have added **"Apple in Mid-air"** real-world experimental results in Figure 12 and "cluttered table"(Figure 13-14), demonstrating the system's ability to grasp objects with vertical variations effectively.
>
> **W6**: Formula 4 confusion
>
> A: We have clarified the meanings of all symbols used in Formula 4 in the revised manuscript.

---

> > ### Author Response · Authors · 2024-11-22
> >
> > **Q1**: Can bounding boxes be scaled or combined?
> >
> > A: Yes, bounding boxes can be scaled according to the size of the object, and combination is also possible. For example, in the newly added "Pour Water" task (see Appendix C), we demonstrate interactions involving multiple objects. For more complex objects, tools like Grounding DINO can provide precise detection of specific parts, such as the handle of a cup, the tissue portion of a tissue box, or a door handle, enabling targeted manipulation. Additionally, SAM can be used for fine-grained segmentation, further enhancing the system's capability to handle intricate objects. These extensions will allow our approach to scale effectively to more complex tasks in future work.
> >
> > **Q2**: Why use fixed cubic spaces?
> >
> > A: Please refer to W3.1 above. While other bounding volumes could be considered, we believe the essence of the analysis remains unchanged. The key objective is to validate the pattern of generalization through incremental spatial expansion, which is effectively captured using fixed cubic spaces.
> >
> > **Q3**: What are the camera setups in real-world experiments?
> >
> > A: There are three cameras mounted on the robot, similar to the Mobile Aloha robot. The left and right cameras are positioned on the wrists of the left and right arms, respectively, providing dynamic first-person perspectives. The center camera, located in the middle of the robot, offers a broader third-person view to reduce partial observability. However, it is often obstructed by the robot's arms, making all three cameras significantly partially observed.
> >
> > As shown in Figure 2, this setup does not increase deployment complexity. It primarily affects the dimensionality of the bounding box representation and policy robustness, offering a realistic and efficient configuration for real-world scenarios.
> >
> > **Q4**: Is the bounding box 2D or 3D?
> >
> > A: The bounding boxes are 2D projections on the image plane, representing the object's spatial location in the image.
> >
> > **Q5**: Can bounding boxes exist in mid-Air?
> >
> > A: Yes, bounding boxes can exist in mid-air. For example, in our newly added experiment, "Bottle in Mid-air," (see Figure 12) the bottle is held by hand while suspended in mid-air, and the policy successfully locates and grasps the object based on the bounding box's position. This demonstrates the system's capability to handle floating objects effectively.
> >
> > **Q6**: What background challenges were faced?
> >
> > A: Background challenges included scenarios such as playing video, dim light, and flickering light, which often caused the detection model to fail to detect objects accurately (see Figure 6 and 7). These visual uncertainties posed significant challenges to the policy. The model's background generalization ability, however, can be attributed to the integration of historical information, multi-camera data, and random masking, in addition to the bounding-box-guided policy. Supporting ablation experiments and detailed explanations are provided in Appendix B.
> >
> > **Q7**: What are the teacher policy's hardware requirements?
> >
> > A: The teacher policy requires minimal hardware resources. Using a single NVIDIA GeForce RTX 4090, we can generate 30,000 simulated trajectories in one day. Our method, ManiBox, emphasizes efficiency and practicality, achieving strong spatial and visual generalization with limited computational resources. For example, training the teacher policy takes only 2 hours, data generation requires 1 day, and training the student policy is completed in 2 minutes on the same GPU.
> >
> > **Q8**: Where are random points placed?
> >
> > A: During testing, random points are generated around the object's center and placed randomly on the table. As shown in Figure 5, these points are carefully distributed to cover every corner and various table heights (i.e., diverse (x, y, z) positions), thoroughly evaluating the policy's spatial generalization ability. Details can be found in Appendix D.3.
> >
> >
> > **Q9**: What are the simulation object types and task settings?
> >
> > A: The simulation uses **apples of varying sizes** as the object type. The task involves **training the robot to grasp an apple**, randomly placed on a liftable table, and transport it to a designated target position. This has been supplemented in Appendix E.1 for further details.
> >
> > **Q10**: How is trajectory quality linked to rewards?
> >
> > A: Trajectory quality is positively correlated with rewards. During data generation in simulation, we use several criteria to filter trajectories:
> >
> > - A trajectory is considered successful if its total return exceeds a threshold (defined as 10 through experiments).
> >
> > - Additional criteria include trajectory length, absence of unexpected terminations, and whether the gripper successfully grasps the object by the end of the trajectory.
> >
> > These measures ensure high-quality data for training. Detailed definitions of the reward functions and their weights are provided in Appendix E.1.4.

---

> > > ### Author Response · Authors · 2024-11-22
> > >
> > > **Q11**: Need for clearer visuals
> > >
> > > A: Thank you for the suggestion. Figure 2 already includes bounding box annotations in simulation screenshots, illustrating the data collection process. To further enhance clarity, we have added additional simulation visuals in Appendix E.1 and Figures 19-21 to help readers, including those without robotics experience, better understand the process.

---

> > > > ### Comment · Reviewer_4x8i · 2024-11-26
> > > >
> > > > Thank you for the thorough clarifications and additional experiments. The new results and explanations address most of my concerns. These updates significantly strengthen the paper, and I have updated my score accordingly.

---

### Official Review · Reviewer_NsAU · 2024-11-04

**Soundness:** 3
**Presentation:** 3
**Contribution:** 3
**Rating:** 8
**Confidence:** 4

**Summary:**

This paper presents a method called ManiBox to train generalizable (to 3D location and object) grasping policies in simulation and successfully transfer them to the real world. The method works by:
1. First, a teacher policy is trained in simulation to grasp objects using Reinforcement Learning
2. A student policy is trained in simulation that uses as observation bounding boxes of the target object from multiple camera streams
3. Bounding boxes provide a low dimensional representation that transfers reasonably well between sim and real. Sim has privileged information so it's easy to get bounding boxes. In real, the YOLO world object detection module is used.

Using this method, the authors show successful sim-to-real transfer and generalization to different objects, backgrounds & 3D locations. The authors also present a study on scaling laws of success rate vs training dataset size.

**Strengths:**

1. Improvement over baseline in the given task of object grasping
2. Insightful study on data scaling laws for robotic object grasping

**Weaknesses:**

1. Lemma 2 is a well known result in 3D computer vision. Hartley, R. and Zisserman, A., 2003. Multiple view geometry in computer vision. Cambridge university press.
2. Limited comparison to baseline. Baseline scores are simply 0 even though there are numerous object-grasping methods (learning & heuristic based) that can grasp objects in different 3D locations.

**Questions:**

1. Line 492-493: Why does vision based ACT get a score of 0? Paper reports that vision based ACT "experiences a visible drop on success rate when spatial range scales up".  Is it because of the visual discrepancy between sim and real?

Prior work has shown strong results of ACT with limited real world data.
Zhao, T.Z., Kumar, V., Levine, S. and Finn, C., 2023. Learning fine-grained bimanual manipulation with low-cost hardware. arXiv preprint arXiv:2304.13705.

2. Line 265: "ensure its dynamics are consistent with the real world". How was it ensured?

---

> ### Author Response · Authors · 2024-11-22
>
> Thank you for your thoughtful review and for recognizing the strengths of our work, including improvements in object grasping tasks, sim-to-real transfer, and the study on data scaling laws. We appreciate your detailed summary and constructive feedback, which motivates us to further refine our work.
>
> **W1**: Clarification on Lemma 2
>
> A: While Lemma 2 may draw from established principles in 3D computer vision, such as those in Hartley and Zisserman (2003), our contribution lies in **its extension to 2D bounding boxes rather than individual points**. Specifically, we compute the 3D object center using multi-view bounding boxes, a novel approach tailored for bounding box-guided robotic manipulation tasks. We have updated the manuscript to provide additional context and references.
>
> **W2**: Limited comparison to baseline
>
> A: Most object-grasping methods are fundamentally different from ours, as our approach focuses on **modeling trajectory data from a simulator** and deploying the student policy on a real robot. Only a few algorithms, such as ACT and Diffusion Policy, have been successfully applied to the Mobile Aloha robot, and both rely on real-world data, which in our experimental setting struggles to generalize across the entire workspace.
>
> Additionally, transferring simulator-generated video data to the real world would require extensive domain adaptation and visual data augmentation, making it **a separate, large-scale effort** beyond the scope of our work. For these reasons, we selected ACT as the sole baseline, as it represents the most relevant approach for comparison in our setup.
>
> Furthermore, the **high success rates** reported in Table 1 highlight the effectiveness of ManiBox. More importantly, this work focuses on uncovering spatial generalization laws, which we believe is a significant contribution to the field.
>
> **Q1**: Why does vision-based ACT get a score of 0?
>
> A: As mentioned earlier, ACT requires training on video-action data. While it can work **under fixed-point conditions** with 100 real-world data samples, this is insufficient for achieving strong spatial generalization. Collecting 20,000 real-world samples, which may be necessary for better generalization, is prohibitively expensive and time-consuming.
>
> When using simulator video data, ACT faces significant **visual discrepancies** between simulation and reality. It can succeed in fixed-point tasks by memorizing joint position trajectories, enabling sim-to-real transfer. However, for larger spatial volumes, ACT fails due to these discrepancies. In contrast, our method is unique in that, beyond using bounding boxes and random masking, it requires no additional sim-to-real visual generalization techniques.
>
> Given ACT's low success rate in our setup, we ensured the fairness of our experiments by thoroughly validating hyperparameters and input configurations at the outset.
>
> **Q2**: "Ensure its dynamics are consistent with the real world". How was it ensured?
>
> A: Ensuring consistency involves two key aspects:
> - **Camera calibration**: We calibrated the intrinsic and extrinsic parameters of the cameras through hand-eye calibration and iterative adjustments, using observed bounding box discrepancies as feedback.
>
> - **Joint position mapping**:
>   - We performed real-to-sim mapping by teleoperating the robot and visually verifying that the joint positions in simulation matched those in the real robot.
>   - For the gripper, as its joint definitions differ between sim and real, we manually measured 10 gripper open-close distances and fitted an approximate linear function to map the gripper joint positions from sim to real.
>
> These steps have been clarified and detailed in the revised manuscript in Appendix E.4.

---

> > ### Comment · Reviewer_NsAU · 2024-11-26
> > **Thank you for the responses**
> >
> > Thank you for the response. I believe the paper can still benefit from a more thorough comparison with baseline methods. As reported, given the difficulty of collecting enough data for ACT to generalize well, I would like to see how the proposed method compared to a more heuristics based approach - e.g. detect bounding box and use 3D geometry to compute object location and send gripper to computed location. Given the limited comparison with baselines, it's difficult to appreciate the advantages of the proposed approach. Hence, I would like to stick to my original rating.

---

> ### Author Response · Authors · 2024-11-29
> **More comparisons with baseline**
>
> Thank you for your continued thoughtful feedback.
>
> In response, in addition to the comparison of ManiBox and ACT in the paper，we have included a more thorough comparison with a **heuristics-based approach**, specifically the **Inverse Kinematics (IK)** algorithm from Isaac Lab, alongside other relevant baselines. We conducted tests using 3 different seeds in the simulator, and the comparison results are as follows:
>
> - **Inverse Kinematics (IK)**: This approach directly uses the ground truth object pose as the target end-effector (eef) pose, then computes the joint positions via IK to move the robot arm to the target. However, we observe that IK performs poorly at the edges of the workspace, especially when objects are near the table surface or at low positions. Examples of IK failures when the table is very low can be seen on this **[anonymous link](https://anonymous.4open.science/r/iclr2025-3C2C/README.md)**. The reason for IK failures is possible:
>     - The robotic arm may approach singular configurations where the Jacobian matrix becomes singular, causing unstable or undefined IK solutions.
>     - Near the workspace edge, numerical inaccuracies may result in imprecise IK solutions. Moreover, approximation methods (e.g., pseudoinverse) may struggle to converge to a valid solution in complex boundary cases.
> - **Key Steps**: In this approach, only key steps of the trajectory are used to model the grasping task.
> - **Without Random Mask**: This variant trains the student policy without random masking for object positioning.
>
> | Method         | Success Rate of Full Space          |
> |----------------|----------------------------|
> | Inverse Kinematics (IK)  | 0.6875 ± 0.0510       |
> | Key Steps      | 0.0625 ± 0.0884       |
> | Without Random Mask   | 0.5833 ± 0.0780       |
> | **Student Policy of ManiBox (Ours)**           | **0.9167 ± 0.0295**       |
>
> Key Insights:
> - **IK Limitations:** As expected, IK struggles in the workspace boundaries due to singularities, which cause failures when the object is positioned at the edges of the table or at low heights. This limitation is inherent to traditional geometric methods like IK, which rely on precise pose knowledge and are prone to errors in challenging environments.
> - **Our RL-Based Approach**: In contrast, **ManiBox (Ours)** utilizes RL to learn robust grasping teacher policies by directly planning the joint positions, enabling successful manipulation across a wide range of object positions. Our method does not rely on explicit pose knowledge and is highly effective even at challenging object locations that cause IK to fail.
>
> We hope this extended comparison addresses your concerns and clearly demonstrates the advantages of our approach over traditional heuristics-based methods, particularly in terms of robustness and generalization. Moreover, as previously mentioned, **We conducted some complex tasks such as "pouring water" (applying our approach to multi-object manipulation tasks) and "grabbing the handle of a cup"(applying our method to the grasping of detailed parts of irregular objects(Appendix C, Figures 10-14, and Supplementary Material)** to demonstrate the scalability of our approach. Thank you for your valuable feedback, and we look forward to your thoughts on these additional experiments.

---

### Official Review · Reviewer_WAZp · 2024-11-09

**Soundness:** 3
**Presentation:** 3
**Contribution:** 3
**Rating:** 6
**Confidence:** 4

**Summary:**

The authors tackle the challenge of spatial generalization and sim-2-real transfer of vision based control policies for robotic manipulation using a teacher-student reinforcement learning framework. The teacher policy, which is trained using privileged information about object location, etc.. is used to generate large scale robot trajectory data which is then used to distill a student policy which achieves sim-2-real transfer to real-world tasks. The authors also shed some light on the amount of robot trajectory data required for a given volume of reachable space of a robot.

**Strengths:**

The paper is well written and easy to read. Aside from a few details that I mention in the questions section, In my opinion the authors provide sufficient details about the experimental set up for an interested reader. The supplementary video provided also supports the paper well.

The challenge being addressed is very relevant to robotic manipulation and the results and methodology presented in the paper are interesting and compelling enough to inspire future research along similar lines.

**Weaknesses:**

1) Pseudo algorithm - In my opinion, the readability of the paper can be improved by including a pseudo algorithm that describes how the teacher is trained --> the criteria for selecting the successful robot trajectory from teacher policy --> The distillation process that generates the student policy could be very useful for a reader.
2) The paper lacks a section on the weaknesses of the current approach. Even in the supplementary video, I dont recall a scenario where the policy fails. I think its important to know some of the failure cases of the method as well.
3) Although this might be common knowledge among RL researchers - more information regarding the reward design could be helpful. Im assuming it involves getting close to the object and then closing fingers, however, this clearly depends on the distance to object and how fast the robot is moving - describing the reward mathematically would be a good addition to the paper.

**Questions:**

The proposed method uses bounding box as object representation for grasping, however clearly bounding box does not capture local surface geometry of the object being manipulated. For example, an odd shape such as banana will not be represented by a bounding box well. Do the authors think that this could be a weakness of this approach? If so, how can it be addressed? I notice that most of the objects being manipulated are spherical/cylindrical in shape which might be easy to grasp, were any other objects used to test the policy?
Another observation I had is that the grasping motion seems consistently similar regardless of the test scenario (be it different object, background), some objects might need grasping from the top, did the teacher policy ever learn this behavior?

Apologize if I missed this, but what is the history length of input observations for the LSTM student policy?

---

> ### Author Response · Authors · 2024-11-22
>
> Thank you for your review and for recognizing the clarity of our presentation, the robustness of our experimental setup, and the potential of our methodology to inspire future research. We also appreciate your acknowledgment of the supplementary video, which supports our work. Your feedback motivates us to refine the manuscript further.
>
> **W1**: Pseudo algorithm
>
> A: To improve readability, we have included a pseudo-code representation in Algorithm 1, titled "Training and Deployment Algorithm of ManiBox." This outlines the teacher policy training, simulation data generation, student policy training, and real-world deployment. We hope this addition enhances clarity and accessibility for readers.
>
> **W2**: Weaknesses and failure cases
>
> A: We acknowledge the following limitations of our approach, which are now discussed in Appendix B.2 of the revised manuscript:
>
> - **Detection errors**: The policy fails when the detection model entirely misidentifies or fails to detect the target object.
>
> - **Bounding box limitations**: Since the policy uses bounding boxes, it may only grasp the center of irregular objects. Future extensions could leverage tools like Grounding DINO for detailed object detection (e.g., grasping specific parts like handles) or SAM for fine-grained segmentation to handle more complex objects.
>
> - **Complex tasks**: For tasks like folding clothes, generating suitable simulator data is challenging, as the policy would struggle to identify appropriate grasp points.
>
> **W3**: Reward design
>
> A: Thank you for your suggestion. We have included a detailed mathematical description of the reward function in Appendix E.1.4 to provide more clarity on its design.
>
> **Q1**: Does the use of bounding boxes limit object representation for irregular shapes?
>
> A: Thank you for this observation. While bounding boxes provide a simple and efficient representation, they may not capture finer details of irregularly shaped objects. One potential enhancement is to **use Grounding DINO for detailed object detection** (Appendix C), such as identifying specific parts like the handle of a cup or a door handle, allowing the policy to target these features. Alternatively, **segmentation results** from SAM could replace bounding boxes for more precise object representation and manipulation.
>
> For objects like bananas, the policy can generalize to elongated shapes if similar examples are included in the simulator data. For instance, in Appendix C, we demonstrate two tasks involving "pouring water" and "Grab the handle of the cup", which includes handling non-spherical objects.
>
> For objects requiring top-down grasps, **the teacher policy could be adapted** during the data generation phase to include such examples. This approach is similar to the pouring task, where modifying the teacher policy's target enables specific actions. Since this work focuses on standard tasks and exploring spatial generalization laws, these extensions are left for future research.
>
> **Q2**: What is the history length of input observations for the LSTM student policy?
>
> A: The LSTM’s history length is effectively infinite. We chose not to set a context length to ensure the student policy has access to the full history of observations. This is particularly important in cases where the detection model fails to identify any bounding boxes midway through a task, allowing the student policy to rely on bounding box data from earlier steps to infer the object's properties.

---

> > ### Comment · Reviewer_WAZp · 2024-11-26
> >
> > Thank you for the detailed revisions made to the initial submission. The new additions have certainly improved the quality of the paper. However, I will be maintaining my original score. I believe that the current problem formulation, particularly the use of bounding boxes, grasping styles that policy exhibits, etc.. imposes inherent limitations on the scope of manipulation challenges that the proposed algorithm can address. While the authors have suggested potential solutions to these limitations, I feel that implementing them would be an entirely new contribution in itself.

---

> ### Author Response · Authors · 2024-11-29
>
> Thank you for your valuable feedback and for acknowledging the improvements in our revised paper. We understand your concerns regarding the use of bounding boxes and the limitations of the grasping policies.
>
> To clarify, the bounding box approach is designed to provide **an efficient way to begin learning spatial generalization**. It is **not meant to limit the scope of manipulation** but rather to create a foundation for future work. In our latest version, we have also compared our method against a more heuristic-based approach (Inverse Kinematics algorithm). Our experiments show that while these heuristic methods are prone to failure, especially at workspace edges, our RL-based approach performs robustly across a wider range of scenarios, as shown by the improved success rates in our experiments:
>
> | Method         | Success Rate of Full Space           |
> |----------------|----------------------------|
> | Inverse Kinematics (IK)  | 0.6875 ± 0.0510       |
> | Key Steps      | 0.0625 ± 0.0884       |
> | Without Random Mask   | 0.5833 ± 0.0780       |
> | **Student Policy of ManiBox (Ours)**           | **0.9167 ± 0.0295**       |
>
> - **Inverse Kinematics (IK)**: This approach directly uses the ground truth object pose as the target end-effector (eef) pose, then computes the joint positions via IK to move the robot arm to the target. However, we observe that IK performs poorly at the edges of the workspace, especially when objects are near the table surface or at low positions. Examples of IK failures when the table is very low can be seen on this **[anonymous link](https://anonymous.4open.science/r/iclr2025-3C2C/README.md)**. The reason for IK failures is possible:
>     - The robotic arm may approach singular configurations where the Jacobian matrix becomes singular, causing unstable or undefined IK solutions.
>     - Near the workspace edge, numerical inaccuracies may result in imprecise IK solutions. Moreover, approximation methods (e.g., pseudoinverse) may struggle to converge to a valid solution in complex boundary cases.
> - **Key Steps**: In this approach, only key steps of the trajectory are used to model the grasping task.
> - **Without Random Mask**: This variant trains the student policy without random masking for object positioning.
>
> Key Insights:
> - **IK Limitations:** As expected, IK struggles in the workspace boundaries due to singularities, which cause failures when the object is positioned at the edges of the table or at low heights. This limitation is inherent to traditional geometric methods like IK, which rely on precise pose knowledge and are prone to errors in challenging environments.
> - **Our RL-Based Approach**: In contrast, **ManiBox (Ours)** utilizes RL to learn robust grasping teacher policies by directly planning the joint positions, enabling successful manipulation across a wide range of object positions. Our method does not rely on explicit pose knowledge and is highly effective even at challenging object locations that cause IK to fail.
>
> We agree that further exploration of more complex object representations and advanced manipulation policies **(We conducted some complex tasks such as "pouring water" (applying our approach to multi-object manipulation tasks) and "grabbing the handle of a cup"(applying our method to the grasping of detailed parts of irregular objects(Appendix C, Figures 10-14, and Supplementary Material))** would be a valuable direction for future work. However, we believe our current approach provides a solid foundation for tackling the challenges of spatial generalization. Thank you again for your constructive suggestions.

---

### Author Response · Authors · 2024-11-22
**Summary Change**

We thank all reviewers for your constructive feedback. Below, we summarize the key updates made to the manuscript which is highlighted in **blue in the revised paper**:

- **Experimental extensions:**
  - Conducted additional experiments on complex tasks such as **"pouring water" (applying our approach to multi-object manipulation tasks)**, **"grabbing the handle of a cup"(applying our method to the grasping of detailed parts of irregular objects)**, and "cluttered table" to validate ManiBox's robustness and scalability (Appendix C, Figures 10-14, and **Supplementary Material**).
  - Expanded on **object diversity and background generalization with ablation studies** (e.g., evaluating the importance of training on entire trajectories and the impact of random masks) and robustness tests using noisy bounding boxes (Appendix B).
  - Introduced new real-world experiments, including "bottle in mid-air" (Figure 12) and "cluttered table" (Figures 13-14), showcasing **generalization to vertical variations and more complex environments**.

- **Discussion of failure cases and limitations:** Included a discussion of ManiBox on its reliance on the preceding visual model for bounding box accuracy and its challenges with irregular object geometries, detection errors, and tasks like folding clothes (Appendix B.2 and Appendix G).

- **Visual and structural improvements**:
  - Included a pseudo-code representation in Algorithm 1, detailing the teacher policy training, simulation data generation, student policy training, and real-world deployment.
  - Included simulation screenshots of key steps decomposition of trajectory for easy understanding of the task (Figure 18).
  - Improved experimental setup explanations and included bounding box annotations in simulation screenshots (Figures 19-21).

- **Clarifications on technical design:**
  - Detailed the Sim2Real Technique (Appendix E.4).
  - Detailed the reward function mathematically (Appendix E.1.4).
  - Provided more details of real-world experiments in order to reflect real-world spatial generalizability, object generalizability, and background generalizability (Appendix D.3).
  - Clarified the reduction of exploration space through the teacher-student framework and privileged information (Section 3.2).

---

### Meta-Review · Area_Chair_Qgja · 2024-12-18

**Metareview:**

The paper presents an approach for robotic grasping through simulation to real training. This a borderline paper with diverging reviews and significant amounts of discussion through the rebuttal and post-rebuttal phase. Hence I have taken a closer look at the paper, the reviews, the author responses, and post-rebuttal discussion. To summarize, the idea of distillation-based sim2real is an interesting idea. The motivation of using such ideas instead of pure imitation learning is quite clear. On the negative side, the major concern is the limited contribution towards novelty and applicability. On the novelty side, the idea of sim2real for grasping and the use of distillation based models for locomotion are quite standard in literature (e.g. DexNet, AnyGrasp). On the applicability side, it is unclear if the produced grasping policies are general enough for diverse real world tasks. For example, recent works like OK-Robot show that grasping in itself can achieve fairly high success rates in real world environments. I hence tend to agree with reviewer rbGJ that the weaknesses of this work outweigh its strengths. Having real world experiments on complex objects in complex real-world scenes would substantially improve the impact of this work in my opinion.

**Additional Comments On Reviewer Discussion:**

On discussion, it remains unclear what the novel contributions of this work really is. The algorithmic ideas have been heavily explored in literature. The application of the technique is limited to a simple object set.

---

### Decision · Program_Chairs · 2025-01-22

Reject